# Impact of Formulation on the Rheological, Textural, and Sensory Properties of Pistachio Spread

**DOI:** 10.3390/foods14234002

**Published:** 2025-11-22

**Authors:** Nazlı Feray Kılıç, Gülten Şekeroğlu, Ahmet Kaya

**Affiliations:** 1Food Engineering Department, Engineering Faculty, Gaziantep University, 27310 Gaziantep, Turkey; nazliferaykilic@gmail.com (N.F.K.); kaya@gantep.edu.tr (A.K.); 2Food Processing Department, Naci Topçuoğlu Vocational School, Gaziantep University, 27600 Gaziantep, Turkey

**Keywords:** *Pistacia vera* L., rheology, anhydrous milk fat, particle size, texture, oil separation, color, nut spreads, sensory attribute

## Abstract

The effects of milk fat (4%, 7%, and 10%) and sugar (24%, 27%, and 30%) content on the physical and sensory properties of pistachio spread (PS) were assessed. Previously, pistachio pastes (PP) were prepared with three particle sizes (large, medium, and small). Small-sized PP was used for nine PS formulations based on the above milk fat and sugar contents. Instrumental texture and color, rheological properties (20–45 °C), and oil separation (4 °C and 25 °C, 9 months of storage) were analyzed in PP and PS. Textural attributes were also evaluated sensorially in PS samples. The oil separation rate in samples stored for 9 months was <1% for PS (4 °C) and >2% for PP and PS (25 °C). The lightness was lower in large-sized PP and higher in PS samples with sugar and milk fat. All PP and PS samples exhibited non-Newtonian behavior with a yield stress. Firmness, spreadability, and adhesiveness were lower in PS samples containing only milk fat. In contrast, they were higher in PS samples containing only sugar. PS samples with 7% milk fat and 27% sugar scored highest for flavor, taste, and acceptability. This study provides the first systematic evaluation of how particle size, milk fat, and sucrose collectively influence the rheological and textural behavior of clean-label Boz Antep pistachio spreads, offering a solid scientific basis for optimizing additive-free nut-based formulations.

## 1. Introduction

Pistachios (*Pistacia vera* L.) are a globally appreciated nut, renowned for their distinct flavor, nutritional benefits, and diverse culinary applications [1,2,3]. According to the most recent report by the International Nut and Dried Fruit Council, global pistachio production reached approximately 1.168 million metric tons (in-shell), marking a 9.1% increase from the previous year [4]. Turkey was among the leading producers, contributing around 383,000 metric tons, which accounts for nearly one-third of the global pistachio supply. Approximately 80% of this production originates from the Southeastern Anatolia region [5].

Among the many cultivars, the “Boz Antep pistachio” holds a distinctive status due to its superior sensory and technological qualities. Boz Antep pistachio is the name given to pistachios harvested before they are fully ripe, approximately one month before the standard harvest time. The term “Boz” originates from the grayish-green outer shell appearance, indicating that the fruit has not yet fully browned. This particular variety is highly prized for its vibrant dark green color, indicating its maturity and chlorophyll content, contributing to its appealing appearance and premium status in confectionery applications [1,6,7]. Additionally, pistachio paste was awarded a Protected Geographical Indication (PGI) designation by the European Commission on 25 June 2025, which mandates that it be produced solely in Gaziantep using early-harvested Boz pistachios, which formally acknowledges it as a legally protected quality characteristic [8]. However, there remains a lack of scientific information on how the distinct compositional characteristics of Boz pistachios impact the physicochemical and textural properties of pistachio-based spreads, despite their cultural and regulatory significance.

Pistachio paste is a preparation made from roasted and crushed pistachio nuts. The process begins with a thermal treatment at approximately 170–190 °C for 8–10 min, followed by grinding to obtain a homogeneous particle size. This paste serves as a key intermediate ingredient for producing pistachio-based products such as pistachio spread. Pistachio spreads are products that contain a minimum of 40% nut ingredients, which may be incorporated in various forms, including whole nuts, nut pieces, paste, or slurry [9]. Pistachio paste is generally preferred in culinary applications because it blends more effectively than pistachio butter. The production technology typically involves a two-step size reduction process. The first phase includes milling with conventional grinding equipment such as a colloid mill, attrition mill, disintegrator, or hammer mill, followed by a secondary homogenization phase to refine the texture [10]. Previous studies have demonstrated that specific drying or storage conditions can influence the composition and quality of pistachio-based products [11].

Pistachio paste consists solely of ground pistachio kernels, whereas pistachio spread includes additional components such as fat and sugar to enhance its structure and flavor. Various stabilization strategies using emulsifiers and oleogelators (e.g., monoglycerides, lecithin, beeswax, and rice bran wax) have been explored to reduce oil separation and improve the storage stability of these semi-solid, oil-continuous systems [12,13,14]. Pistachio products also exhibit complex viscoelastic properties that influence sensory perception, stability, and spreadability. Texture, a multisensory attribute shaped by mechanical and structural characteristics, is essential for consumer acceptance [15,16]. Previous studies have modeled the rheological behavior of pistachio paste using the Herschel–Bulkley flow equation, highlighting its yield stress and shear-thinning behavior [17,18]. Furthermore, particle size distribution, fat–solid interactions, and overall formulation composition strongly influence the rheological and textural properties of nut-based matrices [12,18,19,20,21,22,23,24].

Recent studies increasingly focus on clean-label nut spreads and fat-replacement strategies, including oleogel-based systems and sugar alternatives, reflecting growing industrial interest in healthier spread formulations [25,26,27,28]. Additionally, advances in lipid crystallization and emulsion structuring technologies have improved understanding of fat–solid interactions in nut-based matrices [29]. However, limited research specifically examines pistachio-based systems, particularly regarding the combined roles of particle size, milk fat, and sucrose on rheology and textural attributes, highlighting the need for multidimensional investigations such as the present study.

Previous studies on pistachio-based formulations have primarily investigated single variables such as fat level, particle size, or sugar content in isolation [17,30]. In contrast, the present study systematically examines the combined effects of particle size, milk fat, and sucrose, allowing for a deeper understanding of their interactive influence on rheology and texture. This integrated approach provides unique insight into clean-label pistachio spread development, where formulation must balance texture, stability, and sensory quality without additives. Furthermore, in this study, pistachio spreads were developed using pistachio pastes formulated solely with milk fat and sucrose, without the addition of emulsifiers, stabilizers, or other additives. This clean-label design allows the intrinsic effects of formulation variables on rheological, textural, and sensory properties to be evaluated directly. Given the growing consumer and industrial interest in natural nut-based products, understanding the behavior of additive-free pistachio formulations is essential for the development of high-quality, minimally processed pistachio spreads.

Therefore, this study hypothesized that the physical and sensory properties of pistachio spreads are primarily determined by particle size and the proportions of milk fat and sucrose incorporated into the pistachio paste. To test this hypothesis, the study evaluated (i) the influence of temperature on the rheological behavior of pistachio paste and pistachio spreads, (ii) the textural properties of pistachio spreads as a function of formulation composition, and (iii) the sensory characteristics of the spreads in relation to the proportions of ingredients added. This integrated approach provides a comprehensive understanding of how formulation variables shape the rheological, textural, and sensory attributes of additive-free, clean-label pistachio spreads.

## 2. Materials and Methods

### 2.1. Materials

Boz pistachios, icing sugar, and anhydrous cow’s milk fat were supplied by a local producer in Gaziantep, Turkey.

A laboratory-scale colloid mill (Demirbaş Makina, 2018, Afyonkarahisar, Turkey) was used to produce pistachio paste. Firstly, pistachio nuts were dried in an oven (Heratherm OGH60, Thermo Scientific, Langenselbold, Germany) at 100 °C to achieve a moisture content of less than 3%, thereby preventing clumping during the grinding process. Colloid mills typically use shear force and high-frequency vibrations to reduce the size of food material before forcing the particles to pass through the space between stationary and rotating wheels [31]. The particle size of the material being processed is controlled by varying the distance between the rotor and the stator.

Three pistachio pastes were prepared using a mill with different particle sizes: LP (large), MP (medium), and SP (small). Anhydrous milk fat and powdered sugar were added to the SP pistachio paste to create three different formulations of pistachio spread (PS) (Appendix A) (Appendix A). The mixtures of pistachio spreads were blended in a colloid mill. All pistachio-based samples were prepared, poured into a glass container, and stored in a refrigerator (4.0 ± 1.0 °C) prior to analysis.

### 2.2. Proximate Analysis of Pistachio

The proximate composition of the samples was determined according to the Official Methods of AOAC International [32]. The moisture content of the pistachios was calculated by the difference in weight of an approximately 10 g sample before and after drying at 105.0 ± 1.0 °C for 3 h (Heratherm OGH60, Thermo Scientific, Langenselbold, Germany). Oil in the kernel was extracted with hexane through distillation for 6 h in an automatic extractor (SER 148/6, Velp Scientifica, Usmate (MB), Italy). The solvent was removed in a rotor evaporator at 50 °C for approximately 30 min, and the samples were further dried at 105 °C for 10 min (Heratherm OGH60, Thermo Scientific, Langenselbold, Germany), cooled in a desiccator, and weighed. The crude protein content was analyzed using the Kjeldahl method (AOAC Official Method 991.02) with a nitrogen-to-protein conversion factor of 5.30. The carbohydrate and ash contents were measured according to the standard method [32].

### 2.3. Analysis of Particle Size Distribution

The particle size distribution of the pistachio paste samples was determined using a laser diffraction particle size analyzer (Malvern Mastersizer 2000 E; Malvern Instruments Ltd., Worcestershire, UK). Before measurement, each sample was thoroughly diluted in distilled water (at room temperature) at a 1:10 (*w*/*v*) ratio. Following 1 min of manual shaking, the suspensions were homogenized using a vortex mixer (Heidolph D-91126; Schwabach, Germany) to disperse aggregates and facilitate the effective removal of free oil droplets. The thoroughness of the process was evident by performing all analyses in triplicate and reporting the mean values. Particle size values were expressed as D_10_ (μm), D_50_ (μm), and D_90_ (μm), representing the diameters at which 10%, 50%, and 90% of the total volume of particles were smaller, respectively [17].

### 2.4. Instrumental Color Analysis

Color measurements of the pistachio pastes and spreads were performed utilizing a ColorFlex colorimeter (Model A60-1010-615; Hunter Associates Laboratory Inc., Reston, VA, USA). The color parameters L*, a*, and b* were documented according to the International Commission on Illumination (CIE) color space standard [33]. In this system, the L* value signifies lightness (ranging from 0 = black to 100 = white), a* represents the red–green axis (positive values denote redness; negative values denote greenness), and b* represents the yellow–blue axis (positive values denote yellowness; negative values denote blueness). Before analysis, the equipment was calibrated using standard black and white calibration tiles. Upon inserting the sample into the measuring chamber, measurements were obtained relative to the white standard. Each sample was assessed at no fewer than ten distinct locations to allow for any variability, and the average color values were computed. All measurements were conducted in triplicate.

### 2.5. Oil Separation Rate

A slightly modified protocol of Shakerardekani [17] was followed for the oil separation measurements. Approximately 15 g of pistachio paste samples (LP, MP, SP) and pistachio spread samples (F1–F3, S1–S3, FS1–FS3) were weighed individually into separate centrifuge tubes and heated at 80 °C in a water bath for 30 min. Then the tubes were allowed to cool under running water for 15 min. The tubes were centrifuged with a Merlin Supra centrifuge (Spectra Scientific, Buckinghamshire, UK) for 15 min at 3000× *g* rpm. A Pasteur pipette was used to remove the separated oil. The oil separation rate (OSR) measures the degree of oil phase separation in pistachio pastes and spreads, which is essential for determining their colloidal stability. Additionally, after the samples were centrifuged and the oil phase was separated, they were stored at refrigeration (4.0 ± 1.0 °C) and room temperature (25.0 ± 1.0 °C) conditions for 9 months, simulating shelf storage in a store or household. The amount of oil phase separated at periodic intervals (every week) was measured to investigate the effect of temperature, particle size, and composition on oil stability.

The following formula was used to determine the centrifugal oil–solid OSR:(1)OSR (g/100 g) = (m_1_/m_i_) × 100 where m_1_ is the mass of the separated oil in grams, and m_i_ is the mass of the initial sample in grams.

### 2.6. Rheological Analysis

The rheological characteristics of pistachio pastes and spreads samples were evaluated with a rotational viscometer (Brookfield RVDV-III Digital Viscometer; Brookfield Engineering Laboratories, Middleboro, MA, USA) equipped with a temperature-controlled chamber and operated through Rheocal T 1.0.9 software (Brookfield Engineering Laboratories, Middleboro, MA, USA) for data acquisition and analysis. Measurements were carried out at six distinct temperatures: 20 °C, 25 °C, 30 °C, 35 °C, 40 °C, and 45 °C using spindle No. 27 and a standard sample cup. Shear stress and apparent viscosity were measured for each sample across a shear rate range of 1–50 s^−1^ for a duration of 250 s. The flow characteristics of the samples were expressed using the Herschel–Bulkley equation [18].

The Herschel–Bulkley model equation is as follows:(2)τ = τ_0_ + Kγ *^n^* where

τ = shear stress (Pa);τ_0_ = yield stress (Pa), which is the stress required to initiate flow;K = consistency index (Pa·s), indicating the fluid’s viscosity;*n* = flow behavior index (dimensionless);γ = shear rate (s^−1^).

Furthermore, the relationship between temperature and the consistency coefficient (K) can be accurately described by an Arrhenius-type equation [24]. The model parameters were determined by carefully following the consistency coefficient, K, as a function of temperature using Equations (3) and (4)(3)K = k_0_exp (*E_a_*/RT),(4)ln K = lnk_0_ + (*E_a_*/RT), where

K = consistency coefficient (Pa·s);k_0_ = Arrhenius constant (Pa·s);*E_a_* = activation energy (J/mol), which represents the stability of the system;R = universal gas constant (8.314 J/mol);T = absolute temperature (°K).

### 2.7. Instrumental Textural Analysis

The textural properties of pistachio pastes and spreads were evaluated using a texture analyzer (TA.XT Plus, Stable Micro Systems, Surrey, UK) equipped with a 45 mm diameter spreadability 45° conical probe (P/45C) and 30 kg load cell. Calibration of both force and distance was carried out before each measurement. The test rig comprised a male probe and a female cone fixture. Prior to analysis, the female cone was conditioned at 5 °C for 24 h. After the conditioning period, approximately 8 g of each sample was placed into the female cone, and the male probe was allowed to penetrate the sample at a constant speed of 5 mm/s from a height of 25.0 mm above the surface. Data acquisition and analysis were performed using Exponent software (version 6.1.1.10, Stable Micro Systems, Surrey, UK). Firmness (maximum force used by the probe to penetrate the sample, N), adhesiveness (negative area of the force–time curve, N·s), and spreadability (area under the force–time curve, N·s) were determined from the texture profile curves [25].

### 2.8. Sensory Analysis

A consumer hedonic sensory test was performed to assess the acceptability of pistachio spreads. A total of 20 panelists (10 males and 10 females, aged 20–45 years) participated in the evaluation. Each panelist evaluated three pistachio spreads (FS1, FS2, and FS3) presented monadically in a randomized serving order to minimize bias. Before the evaluation, panelists were informed about the procedure and provided verbal consent. Sensory attributes, including color, flavor, spreadability, flowability, firmness, adhesiveness, and overall acceptability, were scored using a 9-point hedonic scale (1 = extremely poor, 9 = excellent) [9]. Descriptors used in the sensory evaluation trial of pistachio spreads are shown in Appendix A. Evaluations were conducted in individual booths at 25 ± 1 °C under standardized white fluorescent lighting to prevent color perception bias.

For spreadability assessment, samples were served on plain white sandwich bread (untoasted), cut into uniform 3 × 3 cm pieces. To avoid flavor carry-over effects, panelists rinsed their mouths with room-temperature water and unsalted crackers between samples. Samples (approximately 50 mL each) were presented in coded containers labeled with three-digit random numbers to ensure blind evaluation

### 2.9. Statistical Analysis

All experiments were conducted in triplicate, and the results are presented as mean values. Statistical analyses were performed using one-way analysis of variance (ANOVA) with Statgraphics Plus (version 5.1; Statistical Graphics Corp., Herndon, VA, USA). Duncan’s multiple range test was applied to determine significant differences among the samples, and differences were considered statistically significant at *p* < 0.05. In addition, sensory findings were compared with instrumental, rheological, color, and texture measurements to examine consistency between consumer perception and instrumental responses. Relationships were evaluated descriptively by reviewing trends in mean values across formulations.

## 3. Results and Discussion

### 3.1. Composition of Pistachio

The pistachio samples consisted of 46.08 ± 1.52% total oil; 21.57 ± 0.42% protein; 3.74 ± 0.03% moisture; 27.69 ± 1.0%5 carbohydrate; and 1.38 ± 0.02% ash. These values agree with those reported by other authors [11,34,35]. Tsantili et al. [11] conducted studies on the composition of different pistachio varieties.Oil ratios varied from 49.79% to 57.62%, while protein ratios ranged from 19% to 21.8% on a dry basis. This study’s results show that pistachios can be potentially applicable to industrial applications because of their high chemical composition similarity.

### 3.2. Particle Size Distribution of Pistachio Pastes

Table 1 shows the particle sizes in the pistachio paste samples according to the milling cycle. There was a statistically significant difference between pistachio pastes (*p* < 0.05). Considering a cumulative volume of 10% (D_10_), the particle size was reduced by 67% as the milling cycle was increased.

The particle size distribution of the pistachio paste samples varied significantly depending on the grinding intensity used. LP had the coarsest structure, with a D_90_ of 434.80 µm. In contrast, SP had the finest distribution, with a D_90_ of 149.78 µm, which indicates that the system was more homogeneous and smoother. The median particle size (D_50_) ranged from 16.71 μm in LP to 7.18 μm in SP, confirming a progressive reduction in particle coarseness as the particle size class shifted from LP to SP. The D_10_ fraction showed a similar pattern, with SP reaching values close to 1.88 μm, which contributes to making a less gritty product that is easier to spread. Reducing particle size is expected to enhance mouthfeel and increase the interaction of the surface area with the fat matrix, potentially affecting its rheological behavior, oil binding capacity, and sensory perception. The substantial statistical differences (*p* < 0.05) observed among all three samples at D_10_, D_50_, and D_90_ confirm the implemented size reduction treatment effect. As expected, particle size decreased with grinding time. Shakerardekani [17] produced pistachio paste using different colloid mill gap sizes and reported that reducing the mill gap allowed more kernels to enter the space between the discs, resulting in a more uniform paste more quickly, although with a larger particle size. Moreover, similar studies on pistachio pastes in the literature have shown that reducing particle size (from 80 μm to 20 μm) increases paste homogeneity and contributes to improved colloidal stability [17,36].

Due to their fine grindability, Boz pistachios add a unique flavor and an eye-catching appearance to pastries and desserts. They are the key ingredient responsible for baklava’s characteristic golden green color and distinctive aroma [7,8]. Therefore, SP, which has the smallest particle size, was selected for use in this study in formulations containing pistachio paste prepared with milk fat and sugar.

### 3.3. Color of Pistachio Pastes and Spreads

Table 2 shows the color parameters of pistachio pastes and spreads. Significant differences were found between the LP and SP in terms of L*, a*, and b*. Pistachio paste becomes darker as the particle size increases, as evidenced by LP, which has the lowest L* value at 44.34 ± 0.02. Samples with small particles have a large specific surface area and a small diameter, which causes them to scatter more light and appear lighter and more saturated than samples with large particles [37]. The highest b* value at 45.86 ± 0.05, which indicates the greatest yellowness, was observed in sample LP. a* values of pistachio pastes ranged from −3.82 ± 0.05 (LP) to −3.64 ± 0.04 (SP). Shakerardekani [17] reported that there were no significant differences (*p* < 0.05) in the L, a, and b values of pistachio pastes produced using various milling gaps.

The types and amounts of pistachio paste, sugar, and milk fat in the spread formulations typically have an effect on the color parameters. As the amount of milk fat in the product increased, the L* and b* (yellowness) values also increased, which is assumed to be due to the inherent color of milk fat. Different ratios of milk fats affected the a* values of pistachio spreads significantly (*p* < 0.05). In the formulations with added sugar S1, S2, and S3, the color became lighter, while the yellowness value also increased from 44.19 ± 0.12 to 44.72 ± 0.08. The most important parameter of a Boz pistachio is the a* value, which indicates the greenness of the sample [38,39]. According to instrumental color analysis, FS1, FS2, and FS3 showed comparable greenness; no statistically significant differences were observed among them (*p* > 0.05).

### 3.4. Oil Separation Rate of Pistachio Spreads and Pastes

The results of the separated oil values (%) for samples stored at 4 °C and 25 °C are presented in Figure 1, Figure 2 and Figure 3. In samples with different types and ratios of ingredients, less oil separation was observed at 4 °C compared to the samples stored at 25 °C on the same days.

The pistachio paste exhibited reduced stability with increasing particle size. This is because an increasing quantity of larger particles leads to a decrease in the dispersion of solid phase within the oil phase, resulting in decreased stability. The paste became less stable due to a narrower dispersion of the solid phase in the oil phase as the quantity of bigger particles rose.

The oil separation rate and colloidal stability of pistachio paste, when mixed with milk fat and sugar without emulsifiers or stabilizers, can be investigated based on the chemical and physical properties of these constituents. Pistachio paste, which contains approximately 45–60% oil, along with moderate amounts of protein, and carbohydrates, is recognized for its high oil content. The oil phase typically separates due to gravitational forces, which are influenced by the viscosity of the paste and the interactions among its constituents [40].

Research has demonstrated that naturally sourced sugars, when incorporated into emulsions, can alter molecular interactions, resulting in enhanced stability of oil droplets [41]. This discovery is especially pertinent for researchers, food scientists, and product developers focused on food formulation, particularly those engaged with mixtures that incorporate milk fat, sugar, and pistachio paste. Sugar not only imparts sweetness but may also improve emulsion quality by stabilizing protein interactions [42,43]. The incorporation of sugar affects water activity in the mixture, altering its physical dynamics. Sugar acts as a humectant, which may help stabilize the system temporarily by retaining moisture, but it does not inherently prevent oil separation [40]. When sugar is combined with fat, the overall viscosity of the paste may increase; however, it might still facilitate oil migration if the sugar concentration is not optimally balanced. Studies on nut pastes indicate that oil migration can worsen over time, leading to visible separation on the surface of the paste [44].

Milk fat enhances the creaminess and flavor of the mixture but, similar to sugar, does not inherently stabilize it without emulsifiers. The density differences between milk fat and pistachio oil can lead to phase separation, where the denser components float to the top over time, particularly under suboptimal storage conditions [38]. Research shows that oil separation in pastes, including those with native plant oils, typically exhibits initial rapid separation that slows down as the mixture equilibrates [40].

Moreover, environmental factors, such as temperature, significantly affect both oil separation and the overall stability of the mixture. Higher temperatures can increase oil fluidity, exacerbating the separation process. Conversely, lower temperatures may help stabilize the mixture but could negatively affect textural properties if the fats solidify [45]. Faruk Gamlı and Hayoğlu [46] reported that pistachio nut paste stored at 4 °C was more acceptable due to low deteriorative reactions as compared to those stored at 20 °C.

In summary, the oil separation rate and colloidal stability of a blend of pistachio paste, milk fat, and sugar without added emulsifiers or stabilizers are influenced by the inherent properties of the ingredients, environmental conditions, and balance between the fat and aqueous phases. The mixture is susceptible to oil separation over time due to the interactions among the components, their concentrations, and storage conditions.

### 3.5. Rheological Behavior of Pistachio Paste and Spread

The particle size of the pistachio pastes was systematically reduced across formulations, with LP having the largest and SP the smallest particle size. The rheological behaviors of the LP, MP and SP pistachio pastes were analyzed, and the results are presented in Table 3.

The rheological measurements indicate that all pistachio paste formulations exhibited non-Newtonian rheological behavior, which is consistent with the properties typically observed in nut-based spreads and pastes [47]. All formulations exhibited non-Newtonian shear-thinning behavior with yield stress, which aligns with the behavior of high-viscosity food products, such as nut pastes, spreads, many colloidal suspensions, and emulsions [18,48,49,50,51].

Figure 4 shows the shear-thinning behavior observed, where viscosity decreased as shear rate increased. This behavior suggests that the pistachio pastes possessed a structured network that was disrupted at higher shear rates, allowing for easier flow.

The rheological behavior of the pistachio pastes aligns with that in previous studies on nut-based systems, where particle size was found to play a critical role in determining texture and flow properties. For example, a study on hazelnut pastes found that reducing particle size enhanced flow and decreased apparent viscosity [38]. The presence of yield stress in all formulations indicates that the pistachio pastes required a certain threshold of applied stress before they began to flow, a property commonly seen in food pastes and gels [52].

The particle size of the pistachio pastes significantly influenced the rheological and textural properties of the final pistachio paste. As the particle size decreased from LP (largest) to SP (smallest), the consistency and yield stress values increased. Smaller particle sizes typically lead to more compact and cohesive pastes, which exhibit higher viscosity and yield stress, as seen in the transition from LP to SP. This trend is consistent with the findings of Shakerardekani [17], who reported that finer particle dispersions improved the structural stability and viscosity of fat-based pastes. The interaction between fine particles and the fat matrix likely creates more rigid networks, enhancing the yield stress and consistency coefficient values in the paste.

The divergent temperature-dependent behavior between the MP and SP samples can be attributed to differences in their particle packing density and fat–particle interactions. The SP samples, due to their smaller particle size and higher surface area, formed a more cohesive structure that softened more rapidly as the temperature increased. In contrast, the MP samples exhibited a more gradual change due to less compact packing and a lower oil binding capacity. Moreover, Shakerardekani [17] also evaluated the influence of milling parameters on pistachio paste rheology and demonstrated that the Herschel–Bulkley model was the most appropriate to describe its flow behavior, consistent with the modeling approach applied in the present study. Yield stress is a key factor in the spreadability of nut-based products, as it directly affects the ease with which the product can be spread on a surface. The yield stress values decreased as the particle size decreased, suggesting that finer particles enhance a smoother structure with reduced resistance to flow [53]. The consistency coefficient exhibited a similar trend, with SP demonstrating the lowest consistency coefficient value at 3.53 ± 0.01. This suggests that reduced particle size enhances flowability due to minimized internal friction and particle aggregation [17].

The flow behavior index values were well below 1.0, which confirms the shear-thinning nature of the formulations. The increased rheological behavior index values for smaller particle sizes (SP) suggest improved homogeneity and reduced structural rigidity, a pattern that has also been observed in peanut butter formulations [54]. The statistical analysis confirmed significant differences (*p* < 0.05) in yield stress and consistency coefficient values between the different particle sizes, supporting the hypothesis that smaller particles increase paste viscosity and yield stress.

#### 3.5.1. Effect of Milk Fat

The rheological behavior of pistachio mixtures F1, F2, and F3, prepared with increasing milk fat contents (4, 7, and 10%; *w*/*w*), was analyzed. Table 4 summarizes the yield stress, consistency coefficient, and rheological behavior index for each mixture. Similar to the pistachio pastes, all mixtures exhibited non-Newtonian shear-thinning behavior with an apparent yield stress.

The addition of milk fat to the pistachio pastes significantly altered their rheological properties. As observed in the results, the pastes with higher milk fat concentrations exhibited lower viscosity and yield stress values, which is consistent with the behavior of other fat-based pastes. The addition of fat molecules disrupted the continuous network structure formed by proteins and polysaccharides, resulting in a reduction in overall viscosity. Fat serves as a plasticizer in food pastes, reducing the resistance to deformation and making the product softer and more spreadable [55]. This behavior is well-documented in various studies on fat-based emulsions and nut pastes [18,30].

In this study, the impact of milk fat on the rheological and textural properties of pistachio spreads was investigated at various temperatures. Particularly, significant differences were observed at lower temperatures (20 °C and 25 °C), where milk fat remained in a solid state, affecting both the rheological behavior and texture of the spreads. The solidification of milk fat at low temperature led to a more structured matrix in the spread, which restricted its flow behavior. As the temperature increased, the milk fat fully melted, and the spread became more fluid, as evidenced by the reduction in yield stress values. This transition is crucial for applications in food products that require both spreadability and texture consistency, as fat melting plays a key role in determining these properties [46]. As the milk fat content increased, the yield stress values generally decreased. This can be attributed to the lubricating effect of milk fat, which reduces internal friction within the matrix and facilitates flow [56]. F3, with 10% milk fat, exhibited the lowest yield stress (6.05 ± 0.02 Pa at 45 °C), consistent with findings in fat-rich food matrices where higher fat content softens the overall structure [48]. However, at temperatures below 35 °C, the partial crystallization of milk fat introduced structural rigidity, slightly counteracting this softening effect. Milk fat plays a significant role in the rheological properties of pistachio spreads. The solid fat content increases structural rigidity, leading to higher yield stress and consistency coefficients at 20 °C and 25 °C. This effect diminishes at high temperatures (above 35 °C) as milk fat melting increases, which improves rheological behavior and reduces resistance to deformation [52]. Similar behavior has been reported in studies on fat-containing emulsions, where solid fat content directly impacts texture and viscoelastic properties [48]. For instance, at 20 °C, the yield stress of F3 (10.51 ± 0.16 Pa) was relatively higher compared to its value at 35 °C (6.11 ± 0.07 Pa) or above, suggesting the dual influence of fat crystallization and concentration.

The consistency coefficient followed a similar trend, decreasing with increasing milk fat content. Higher fat content enhanced rheological behavior by disrupting the rigid network of pistachio particles and sucrose. This behavior was most evident at higher temperatures (above 35 °C), where milk fat was fully melted, creating a more homogenous and fluid system [57].

The rheological behavior index values show an increasing trend with milk fat content and were well below 1.0, confirming the shear-thinning nature of the formulations. This indicates that higher milk fat content contributed to a more fluid-like behavior, reducing the non-linearity of the shear-thinning response. The trend suggests that milk fat’s role as a plasticizer becomes more prominent as its concentration increases [38].

Statistical analysis exhibited significant differences (*p* < 0.05) in yield stress and consistency coefficient values between different milk fat concentrations. Spreads with higher milk fat exhibited significantly lower viscosity and yield stress compared to those with lower milk fat, highlighting the importance of fat as a rheological modifier [47].

#### 3.5.2. Effect of Sugar

Table 5 presents the rheological parameters of pistachio mixtures, S1, S2 and S3, prepared with increasing sucrose concentrations (24, 27, and 30%; *w*/*w*, respectively). It should be noted that below 30–35 °C, especially in formulations with higher sugar content, sucrose crystallizes at lower temperatures, making it impossible to obtain accurate flow data. This situation has led to partial solidification and the loss of observable flow behavior, making it impossible to accurately characterize the rheological properties in this temperature range.

The sucrose content directly influenced the yield stress, consistency coefficient, and rheological behavior index values, demonstrating the critical role of solid content in the rheological properties of these systems. This is supported by Faruk Gamlı and Hayoğlu [46], who observed that sugar increases the rigidity of nut-based pastes by forming a structured matrix within the fat phase. Higher sucrose content also led to reduced spreadability; the ability of sucrose to bind moisture and increase paste thickness also contributed to this effect [58].

The consistency coefficient also showed an increasing trend with sucrose content, as expected. S3 demonstrated the highest values at 25.16 ± 0.62, reflecting its more viscous nature. Sucrose acted as a filler within the matrix, increasing the apparent viscosity and energy required for deformation. These interactive effects align with those observed by Rao [52], who suggested that fat serves to soften pastes, while sugar increases their consistency. At higher temperatures, the consistency coefficients decreased significantly *p*< 0.05) due to the melting of milk fat, which offset some of the viscosity contributed by sucrose. These results are similar to those found in studies of other food systems, such as peanut butter, where the combination of fat and sugar results in pastes with varying textural properties [55].

The rheological behavior index values decreased with increasing sucrose content. This indicates a more pronounced shear-thinning behavior for formulations with higher sucrose concentrations. The lower values in S3 (0.581 ± 0.010 and 0.572 ± 0.010) reflect the strong interactions between solid sucrose particles and the pistachio paste matrix, which resist flow under low shear conditions.

These results are consistent with previous studies on sugar-rich food systems. Emadzadeh et al. [47] reported that higher sucrose content significantly increases yield stress and viscosity due to its role in forming a dense and rigid matrix. Similarly, studies on confectionery products have highlighted the interplay between sugar content and fat crystallization in determining rheological behavior [54].

The statistical analysis revealed significant differences (*p* < 0.05) in the viscosity and yield stress of pastes with varying sucrose concentrations. Pastes with higher sucrose content exhibited higher yield stress and viscosity, confirming the impact of sucrose on the textural properties of the pastes.

#### 3.5.3. Combined Effect of Milk Fat and Sugar

The rheological behavior of the FS1, FS2, and FS3 pistachio spreads, prepared with varying pistachio paste (SP) formulations, milk fat, and sucrose, was analyzed, and the results are presented in Table 6. The combined effect of milk fat and sucrose in the pistachio paste formulations resulted in complex changes in the rheological properties. FS1 (4% milk fat and 30% sucrose) exhibited higher consistency, higher yield stress, and lower rheological behavior index values compared to FS2 (7% milk fat and 27% sucrose) and FS3 (10% milk fat and 24% sucrose). The increase in fat content tended to soften the paste by reducing yield stress at higher temperatures, but sucrose counterbalanced this effect by enhancing viscosity at lower temperatures. The combined interactions of fat and sugar have been shown to produce desirable texture properties, such as increased firmness and reduced spreadability at lower temperatures [51]. These interactions result in a balanced formulation where both spreadability and firmness are optimized for consumer preferences.

Yield stress showed an increasing trend with higher sucrose concentrations. FS1, containing 30% sucrose, exhibited the highest yield stress across all temperatures. This is due to the increased volume fraction of solid particles and the formation of a gel-like structure within the paste, which enhances network rigidity and requires higher stress to initiate flow [59]. This trend aligns with findings in similar systems where sugar crystals contribute significantly to the mechanical strength of the structure [9]. At temperatures below 35 °C, this effect was compounded by the partial crystallization of milk fat, further enhancing rigidity and yield stress, particularly in FS3.

Measurements were conducted at temperatures ranging from 20 °C to 45 °C. Across all formulations, yield stress and the consistency coefficient decreased with increasing temperature, as expected due to the softening of lipid components and increased molecular mobility. The yield stress values exhibited strong dependence on the formulations. FS1 (66% PP3, 4% milk fat, 30% sucrose) exhibited the highest yield stress across all temperatures due to the high sucrose concentration and low milk fat content. Conversely, FS3 (66% SP, 10% milk fat, 24% sucrose) had the lowest yield stress, reflecting the softening effect of increased milk fat and the reduced rigidity from lower sucrose content [50].

The consistency coefficient followed a similar pattern to yield stress. FS1 demonstrated the highest consistency coefficient values at 19.01 ± 0.21, suggesting a more viscous and rigid structure, while FS3 had the lowest consistency coefficient values at 2.03 ± 0.04 due to its relatively softer and more fluid composition. These results highlight the balance between sucrose’s solidifying effect and milk fat’s plasticizing effect [47].

Significant differences (*p* < 0.05) were observed between the different fat–sucrose formulations. Spreads with higher fat and lower sucrose concentrations exhibited significantly lower viscosity, whereas pastes with higher sucrose and lower fat concentrations exhibited higher viscosity and yield stress [48].

As the temperature increased, a general decrease in both consistency coefficient and yield stress was observed, as shown in Table 6. This trend aligns with the findings of several studies suggesting that temperature has a significant effect on the rheological properties of food products, particularly in emulsions and fat-based systems [52]. As temperature increases, the molecular motion within the paste becomes more pronounced, leading to a reduction in viscosity. Additionally, the melting of milk fat further reduced the product’s resistance to flow.

Sucrose, on the other hand, had a contrasting effect. As sucrose concentration increased, both consistency coefficient and yield stress values increased (S1, S2, and S3). This suggests that sucrose contributes to the formation of a more rigid network structure, increasing the resistance to flow. Similar effects have been observed in other food systems, where sugar acts as a structure-building agent in emulsions and gels, thereby increasing viscosity and firmness [47]. The high yield stress observed in the pistachio–sucrose formulations suggests that the presence of sucrose increases the structural integrity of the paste, making it firmer and more difficult to spread. Sucrose also played a role in modulating the rheological properties of the pistachio pastes. The effect of sucrose was observed in the increase in consistency and yield stress at higher sucrose concentrations, especially at lower temperatures. For example, pistachio spreads containing 30% sucrose (FS1) were thicker and more resistant to flow at 20 °C and 25 °C, likely due to the increased interaction between sucrose and the fat phase. At these temperatures, the pastes exhibited high viscosity and higher yield stress values, which could be attributed to both the solidified fat and the thickening effect of sucrose in the system.

As temperature increased, the viscosity decreased, and the pastes became more fluid, consistent with the melting of milk fat and the dissolution of sucrose, leading to a more uniform and less resistant paste. The effect of sucrose concentration on viscosity is well-documented in other food systems, where higher sugar concentrations lead to thicker pastes and increased yield stress [50]. However, at higher temperatures, the decrease in viscosity was more pronounced, reflecting the dominant effect of milk fat melting on the overall rheology. Therefore, at 20 °C and 25 °C, the fat remained in a semi-solid or crystalline form, resulting in a marked increase in viscosity and a decrease in the flowability of the pastes, as shown in Table 6. This behavior was most noticeable in formulations with higher milk fat content, especially those containing 10% milk fat (FS3). At these temperatures, the presence of solidified milk fat contributes to the overall stiffness of the paste, which limits its ability to flow and results in increased resistance to shear stress. This was evident in the rheological measurements, where the pastes exhibited higher yield stress values, particularly at 20 °C (16.78 ± 0.41 Pa) and 25 °C (13.40 ± 0.14 Pa), indicating that the milk fat had not fully melted.

The Arrhenius parameters, including activation energy (*Ea*), were determined for all formulations based on the temperature dependence of yield stress. These parameters provide insights into the energy required to overcome resistance to flow as the temperature changes. The results are summarized in Table 7.

The pistachio pastes’ activation energy (*E_a_*) values slightly decreased as the particle size decreased. The largest particle size, LP, also had the highest (*E_a_*) (32.1 kJ/mol), suggesting that its coarser and more rigid microstructure requires more energy to start flow. However, the *E_a_* values for MP and SP, which had progressively smaller particle sizes, were slightly lower (31.8 kJ/mol), indicating that finer particles make molecular mobility and flow easier in a thermal environment. Moreover, SP had the lowest pre-exponential factor (k_0_) values (2.1 × 10^−5^), followed by MP (2.4 × 10^−5^) and LP (2.8 × 10^−5^), indicating that better flowability is a result of smaller particle size. Strong temperature-dependent variation across particle size treatments was indicated by the excellent Arrhenius fit (r2 ≥ 0.986) of pistachio pastes.

The activation energy (*E_a_*) values increased with sucrose concentration, rising from 15.3 kJ/mol at 24% sugar to 18.4 kJ/mol at 27% and 18.9 kJ/mol at 30% sucrose. This trend clearly indicates that formulations with higher sucrose content exhibited greater temperature sensitivity, affecting their viscosity, which confirms that sucrose strengthened the temperature dependence of flow behavior. Pastes with higher fat content exhibited lower activation energies, consistent with findings by Emadzadeh et al. [51], which indicate that fat reduces temperature sensitivity in emulsions. The reduced temperature dependence in fat-rich pastes suggests that fat stabilizes the emulsion, maintaining a consistent texture across a broader temperature range. This behavior may also be affected by variations in the fatty acid compositions of pistachio oil and milk fat. Pistachio oil is primarily abundant in unsaturated fatty acids, while milk fat comprises a greater percentage of saturated fatty acids, which typically create more solid crystalline structures. While the fatty acid content was not examined in this investigation, saturated lipids typically enhance thermal stability in fat-based matrices; therefore, this mechanism may have contributed to the diminished temperature sensitivity shown in the milk-fat-enriched samples.

The activation energies determined for the pistachio pastes in this study are in line with those reported for similar food systems. For example, in chocolate spreads, activation energies typically range from 20 to 40 kJ/mol, depending on the fat and sugar contents [52]. Similarly, the observed relationship between sucrose content, milk fat crystallization, and activation energy is consistent with findings from studies on nut-based and fat-rich spreads [17].

Temperature had a significant effect on how the pistachio pastes flowed. As anticipated, both the yield stress and consistency coefficients diminished with rising temperature, aligning with the conventional behavior of non-Newtonian food systems, wherein thermal input diminishes internal structural resistance and facilitates flow. These results agree with the findings of Cruz et al. [60], who indicated that increased temperature diminishes the internal resistance of pastes, thereby improving their flowability and spreadability.

The temperature-dependent decline in the flow behavior index (*n*) indicates stronger shear-thinning behavior at lower temperatures (20–30 °C) and a gradual shift toward more Newtonian flow at higher temperatures (40–45 °C). The rheological behavior of the pistachio spreads can be explained by the combined effects of particle morphology, fat crystallization, and sucrose-mediated solid interactions. Finer particle size increased yield stress (τ_0_) and consistency due to tighter particle packing and larger surface area, leading to a more continuous and cohesive structural network. This observation aligns with particle dispersion theory, where smaller particles enhance interparticle friction and structural rigidity.

Milk fat primarily acted as a softening and lubricating phase. According to fat crystal network theory, the proportion and morphology of solid fat govern the firmness and temperature sensitivity of fat-based systems. As noted by Małkowska et al. [61], milk fat crystallization involves several polymorphic forms (α, β′, β), whose transitions influence the mechanical stability of the lipid matrix. At lower temperatures, higher solid fat content reinforces the network and increases τ_0_, while at temperatures above 40 °C, crystal melting and polymorphic conversion weaken the structure, thereby reducing yield stress and viscosity. This behavior reflects the dual role of solid milk fat crystals: at low temperatures, crystal structures act as rigid fillers that increase network strength and limit particle mobility, while at higher temperatures, melted fat contributes to lubrication and reduces internal friction, facilitating flow and spreadability. Such polymorphism-driven transitions are characteristic of lipid-rich colloidal systems.

Conversely, sucrose strengthened the solid framework by promoting crystalline bridging and particle adhesion, which increased τ_0_ and decreased spreadability. Because sucrose remains crystalline within the tested temperature range (20–45 °C), it contributes rigidity without undergoing glass-transition softening. It also contributes to microstructural hardness by forming a crystalline network within the oil continuous phase. This increases interparticle cohesion and reduces molecular mobility, thus providing higher hardness and energy requirements during deformation. This partly explains the formation of a regular sucrose matrix, higher viscosity, and lower spreadability in high-sucrose formulations.

Overall, these findings indicate that smaller particle size and higher sugar concentration reinforce the solid matrix, whereas higher milk fat levels promote flow by disrupting crystal–particle interactions. The textural and rheological properties of pistachio spreads are therefore determined by the balance between these opposing mechanisms.

Additionally, a statistical analysis of the K values obtained from rheological analyses of the samples at various temperatures was conducted (Table A1). Significant statistical differences were found at all temperatures studied among samples FS1, FS2, and FS3 (*p* < 0.05). Similarly, significant differences were also found between the K values of products with different particle sizes (LP, MP, and SP) depending on temperature (*p* < 0.05). Apart from being statistically significant, the observed differences in K values have technological significance. When K values are higher, the structures are firmer, and the spreadability is lower. In contrast, lower K values favor softer matrices that are easier to spread, which is relevant for processing ability and consumer texture perception.

The determined activation energy (*E_a_*) values indicate that the rheological behavior of the pistachio pastes was highly sensitive to temperature changes. The higher activation energy in LP reflects a more temperature-dependent rheological behavior, likely due to the increased influence of sucrose and the partial crystallization of milk fat at lower temperatures. As temperature increased, milk fat fully melted, reducing the resistance to flow and lowering yield stress and viscosity [53]. In contrast, the pistachio pastes with sugar-based formulations (S1, S2, and S3), with sucrose contents increasing from 24% in S1 to 30% in S3, exhibited progressively higher *E_a_* values, indicating that higher sucrose levels promote the formation of a more rigid and structured network that requires greater energy to initiate flow. This behavior is in agreement with previous findings on sugar-rich nut-based systems, where sucrose-driven solid–liquid interactions and microcrystalline network formation increased the temperature sensitivity of viscosity and delayed the onset of flow under thermal excitation. Emadzadeh et al. [50] found that the addition of sucrose at high concentrations increased the viscosity and stiffness of food pastes, leading to higher activation energy values.

These activation energy (*E_a_*) values are also consistent with those reported in the literature for similar fat-based food systems. Rabadán et al. [56] found that fat-based pastes exhibited a strong temperature dependency, with higher activation energies corresponding to pastes with higher fat content. This behavior is attributed to the increased energy required to overcome intermolecular forces in the solid phase of fat, leading to significant changes in rheological properties as the fat melts. Beyond total fat content, the fatty acid composition of the lipid matrix is a critical contributor to this response. Specifically, pistachio oil is rich in polyunsaturated fatty acids, which typically form more fluid networks, while milk fat contains a higher proportion of saturated fatty acids, promoting more rigid and stable crystalline structures. Indeed, Małkowska et al. [61] reported that fatty acid content profoundly affects the quality and consistency of butter, often termed as firmness/hardness and spreadability. Although fatty acid profiles were not analyzed in this study, these compositional differences in the lipid matrix partially explain the distinct, temperature-dependent rheological behavior observed in formulations containing milk fat.

### 3.6. Textural Properties of Pistachio Paste and Spread

The textural properties of pistachio paste and spread were analyzed, and the calculated values are presented in Table 8. The sucrose-containing formulations (S1–S3) exhibited the highest firmness values, ranging between 5.05 N and 5.33 N. In contrast, the milk-fat formulations (F1–F3) displayed the lowest firmness values (between 1.79 N and 1.02 N). As particle size decreased, firmness values declined from 2.69 N to 2.36 N, indicating a softer structure with finer particles. Among all samples, S3, which contained the highest sucrose level (30%), demonstrated the greatest firmness (5.33 N). Firmness, assessed as the peak value in the spreadability analysis, indicated that pistachio pastes with higher sucrose concentrations (S1, S2, and S3) were significantly firmer. This finding aligns with previous research by Dubost et al. [54], who discovered that sucrose enhances the firmness of pastes by contributing to a more structured network. The increased firmness can be attributed to the ability of sucrose to form hydrogen bonds with other components in the paste, creating a more rigid structure [62].

As shown in Table 8, spreadability was significantly affected by both milk fat and sucrose content. Higher fat content led to better spreadability, as it decreased yield stress and enhanced fluidity [46]. The lubricating effect of milk fat was most pronounced in the SP sample, where the paste was smooth and easily spreadable. In contrast, higher sucrose content reduced spreadability, as the paste became thicker and more resistant to shear, supporting findings by Emadzadeh et al. [55]. The crystallization of milk fat also influenced spreadability. At lower temperatures (below 35 °C), the crystallization of milk fat increased yield stress, reducing spreadability, which is in line with previous studies [56]. This effect was particularly notable in the pastes stored at 20 °C and 25 °C, where pastes became firmer as the fat crystallized.

The spreadability of the pistachio pastes was significantly influenced by both milk fat content and sucrose concentration. FS3, which had the highest milk fat (10%) and the lowest sucrose content (24%), exhibited the greatest spreadability. This can be attributed to the lubricating and plasticizing effect of milk fat, which softened the paste and reduced resistance to spreading [63]. In contrast, FS1, with the lowest milk fat (4%) and highest sucrose content (30%), exhibited the least spreadability due to its firmer, more rigid texture. As expected, FS2 (7% milk fat, 27% sucrose) exhibited intermediate spreadability, confirming the influence of both components on the overall spreadability. The results demonstrate a clear trend where higher milk fat content facilitates easier spreading, while higher sucrose concentration contributes to greater resistance to deformation, making the paste more difficult to spread.

The work of shear, representing the energy required to spread the paste, was also significantly higher for the pistachio–sucrose formulations. This result highlights the inverse relationship between firmness and work of shear, as firmer pastes typically require more energy to spread. Conversely, pastes with higher milk fat concentrations (F1, F2, and F3) exhibited lower firmness and work of shear values, making them easier to spread. These findings are consistent with the work of Emadzadeh et al. [55], who exhibited that increasing fat content results in softer, more spreadable pastes with reduced resistance to shear. The spreadability of nut-based pastes is often governed by the balance between fat and sugar contents, as seen in other studies on nut butters and spreads. For example, a study on hazelnut spreads found that increasing fat content improved spreadability, while higher sugar content decreased it [64]. Similarly, in chocolate pastes, fat content was identified as a key factor in determining firmness and spreadability, depending on the nature and amount of the lipids used for their preparation [65]. The statistical analysis exhibited significant differences (*p* < 0.05) in spreadability values based on fat and sucrose concentrations. Higher fat concentrations resulted in significantly better spreadability, while higher sucrose concentrations led to reduced spreadability.

The adhesiveness values make it evident how the formulation parameters affected the spreads’ stickiness and ability to form residues. Adhesiveness gradually decreased from −0.667 to −0.538 N·s. in the particle size series (LP, MP, SP), suggesting that smaller particles were less sticky, most likely due to reduced inter-particle friction and clearer oral perception. The lubricating/plasticizing functions of fat were confirmed by a similar reduction trend observed as the milk fat content increased (F1–F3), where adhesiveness values dropped from −0.401 to −0.224 N·s. This implies a less sticky and smoother mouth feel, which is often associated with improved sensory acceptance. In contrast, with adhesiveness values ranging from −1.272 to −1.300 N·s, the sucrose series (S1–S3) showed the highest stickiness and residue formation. Since higher sugar levels are known to enhance capillary forces and structural cohesion, resulting in greater adhesive resistance, this behavior aligns with results in the existing literature. The behavior of the combined sugar–fat formulations (FS1–FS3) was balanced and intermediate. Because of the predominant effect of fat, FS1 showed comparatively high adhesiveness (−0.796 N·s), while FS3 showed noticeably lower stickiness (−0.352 N·s). These findings unequivocally show that while sufficient milk fat can counteract the tendency of sucrose to increase adhesiveness, excessive levels may also reduce desirable textural resistance. This formulation-dependent behavior in adhesiveness is consistent with the broader understanding of fat–solid and sugar–solid interactions in spreadable food systems.

The results of the present study are consistent with recent investigations, highlighting the role of lipid network structuring and sugar–fat interactions in determining the spreadability, stability, and viscoelastic properties of nut-based spreads [64,66]. In agreement with these outcomes, research on hazelnut and chocolate spreads has demonstrated that fat composition and microstructural network development significantly influence firmness, flow behavior, and melting characteristics [26,27]. Moreover, recent work has emphasized the importance of fat crystallization mechanisms and the distinct functional behavior of polyunsaturated versus saturated lipids in shaping textural stability and consumer perception in spreadable systems [28,29]. Collectively, these findings underscore the relevance of the present research and reflect the growing interest in clean-label, naturally structured nut-based spreads in both scientific and industrial contexts.

### 3.7. Sensory Properties of Pistachio Paste and Spread

Figure 5 illustrates the radar plot of mean hedonic scores for the sensory attributes, allowing a comparative visualization of consumer acceptance patterns across the pistachio spread samples. The sensory spider plot provides a clear visual representation of FS2’s superiority over FS1 and FS3 in almost all of the traits examined. FS2’s top rankings in flavor, taste, color, spreadability, and overall acceptability indicate strong consumer preference. Its optimal stickiness contributes to its excellent spreadability and mouthfeel, further enhancing its appeal. While FS3 exhibited slightly better flowability and less stiffness, FS2 emerged as the most balanced in terms of texture, taste, and handling attributes. This balance makes FS2 the formulation most preferred in sensory evaluation. FS1, with lower scores across most qualities, may not be as appealing to consumers.

Figure 5 demonstrates that FS2 (66% pistachio paste (SP), 7% milk fat, and 27% sugar) received the highest mean scores for sensory attributes (taste, flavor, color, spreadability, and overall acceptability). Therefore, this was considered the most acceptable spread, being rated from very good to excellent. As supported by both instrumental and sensory evaluations, FS2 demonstrated the most desirable overall quality. The instrumental texture results show that FS3 was the least hard and sticky, but the sensory panel liked FS2 better because it had a more balanced texture—neither too firm like FS1 nor too soft and flowable like FS3. FS2 had the best mix of moderate firmness, acceptable adhesiveness, and decent spreadability. This led to the top marks in taste, flavor, spreadability, and overall acceptance during sensory evaluation. These results show that consumers prefer a moderate texture profile over one that is very soft or very firm.

The findings from this study have significant implications for the development of pistachio-based spreads. The ability to control the spreadability, firmness, and texture of the paste through the manipulation of milk fat and sucrose contents allows for the creation of customized pistachio spreads with desired sensory characteristics. For example, a lower sucrose content and higher milk fat would result in a smoother, more spreadable paste, suitable for applications requiring easy spreadability (e.g., on bread). On the other hand, higher sucrose content and lower milk fat would result in a firmer paste, ideal for confectionery uses or as a topping [30,64]. Furthermore, the rheological and textural properties of pistachio pastes could be optimized for specific consumer preferences by adjusting the formulation to meet varying needs for sweetness, consistency, and spreadability. The findings of this study can also inform processing techniques, such as the control of temperature during production, to achieve the desired product characteristics.

The combined manipulation of the sucrose and milk fat levels had a significant impact on the structural and sensory quality of the pistachio spreads, as the instrumental texture results of the FS series clearly showed (*p* < 0.05). Firmness and work of shear values steadily declined as the milk fat content increased from FS1 to FS3, suggesting a matrix that was becoming more lubricated and plasticized. The sensory evaluation showed that FS3 was the most fluid and easily deformable formulation, with a flowability score that increased significantly from 6.94 in FS1 to 8.06 in FS3. This supports known rheology–sensory relationships, as previously documented for spreads made with pistachios and other nuts, where a higher fat content improves spreadability and oral melting behavior while lowering particle–particle friction [13,21,22]. Nevertheless, despite having the highest flowability, FS3’s overall acceptability (7.25) was lower than that of FS2 (8.06), suggesting that excessive fluidity and a lack of structural definition could undermine consumers’ perceptions of premium texture, particularly in conventional pistachio-based applications.

FS1, which had the lowest fat content, showed lower spreadability scores (6.82–6.94 range) and higher mouth adhesiveness in sensory evaluation. Consistently, instrumental texture measurements indicated greater negative adhesiveness and higher stiffness for this formulation, supporting the observed sensory perception trend. Perceived stickiness was considerably higher for sensory qualities, such as mouth adhesiveness (6.94) and adhesiveness to the spoon (7.29), as compared to FS2. Similar to this tendency, insufficient lubrication leads to structural resistance and the formation of oral residue in high-solid nut butters [12,14]. Although scores above seven imply a “good” perception, FS1 lacked the harmony, richness, smoothness, and softness required for increased hedonic appeal.

FS2 consistently stood out as the best formulation, with the highest overall acceptability (8.06), a delicious flavor (7.88), acceptable spreadability (7.53), and a perfect balance of mouth feel. It also exhibited intermediate stiffness and moderate adhesiveness, as indicated by instrumental data. This suggests that FS2 achieved the critical sensory–rheological equilibrium zone, wherein sufficient plasticization by milk fat reduces structural resistance without compromising body, and sugar provides body and taste perception without creating excessive hardness or stickiness.

The significantly higher flavor and taste scores in FS2 suggest that the fat–sugar ratio may have optimized flavor release kinetics, enhancing both sweetness perception and lipid–aroma solubilization. These findings indicate a consistent alignment between instrumental texture assessments and sensory evaluations, particularly in terms of spreadability, adhesiveness, and flowability. Notably, FS3 demonstrated that the highest instrumental fluidity does not necessarily translate into the greatest consumer preference, as excessive softness may compromise textural satisfaction. Sensory acceptance was maximized when structural integrity, lubricity, and flavor perception were simultaneously balanced, a condition most effectively achieved by FS2. All formulations, made without emulsifiers, stabilizers, or commercial additives, achieved high sensory scores (≥ 6.5), demonstrating that clean-label pistachio spreads can be successfully developed using only natural macronutrient modulation. Consistent with recent clean-label nut-spread formulation trends, the natural fat–sugar balance and particle structure played key roles in sensory acceptance [25,26].

## 4. Conclusions

The present study highlights the intricate relationship between particle size, milk fat, and sucrose in influencing the rheological properties and spreadability of pistachio spreads. Finer particle sizes increased viscosity and yield stress, while milk fat reduced these properties, particularly at higher temperatures. In contrast, sucrose increased viscosity and decreased spreadability by reinforcing the structural network and enhancing solid–solid interaction. Furthermore, the findings emphasize the importance of controlling thermal conditions, fat phase transitions, and sugar interactions to produce high-quality pistachio spreads with enhanced sensory attributes. The results provide useful guidance for balancing milk fat and sucrose levels to achieve desirable spreadability and stability.

This research provides the first integrated evaluation of multi-factor interactions in Boz Antep pistachio spreads under clean-label conditions, offering a mechanistic understanding that links formulation variables to rheological and textural performance. These insights contribute to the scientific foundation for developing naturally structured, additive-free nut spreads with improved consumer appeal and industrial applicability. In order to obtain healthy and sustainable pistachio spreads, future research should focus on finding natural alternatives to sugar and plant-based substitutes for dairy fat. New formulations should not compromise textural properties or consumer acceptability. Additionally, comprehensive studies that utilize microstructural imaging and shelf-life evaluations would enhance our understanding of how these factors influence consumer acceptance and product stability over time. Furthermore, investigating the impact of fluctuating storage temperatures and real-life distribution environments would further strengthen the industrial relevance of this research. From an industrial perspective, these results can be used to develop guidelines for formulation and process design. For example, maintaining medium particle sizes and moderate milk fat levels (6–8%) can make the product easier to spread and prevent the oil from separating. In contrast, adjusting the sucrose concentration (25–28%) can give the product acceptable structural strength without being too sticky. The stated temperature range (20–45 °C) also provides a practical way to manage the flow of materials when combining, filling, and storing. The results provide actionable insights for creating standardized, clean-label pistachio spreads that exhibit consistent rheological properties and a texture acceptable to consumers.

## Figures and Tables

**Figure 1 foods-14-04002-f001:**
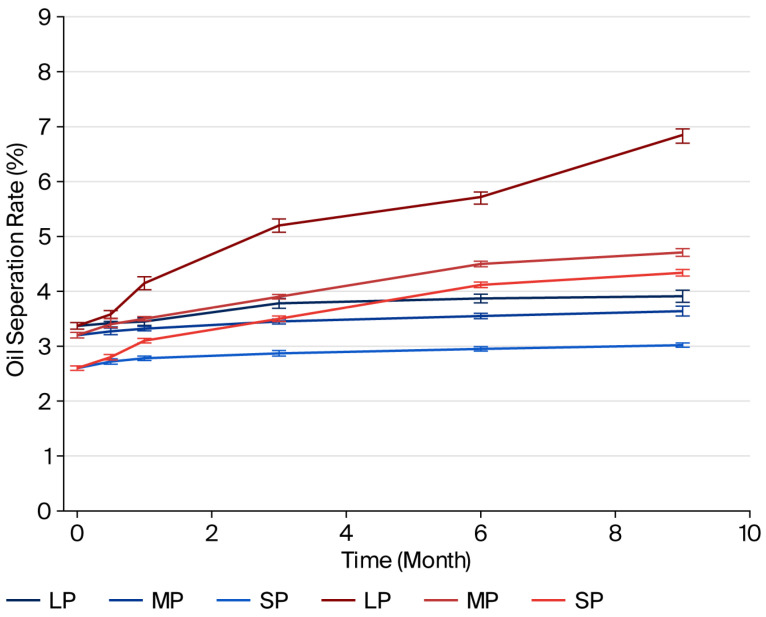
Oil separation rates of pistachio pastes at 4 °C (blue lines) and 25 °C (red lines) (the length of the error bars represents the standard error of the mean).

**Figure 2 foods-14-04002-f002:**
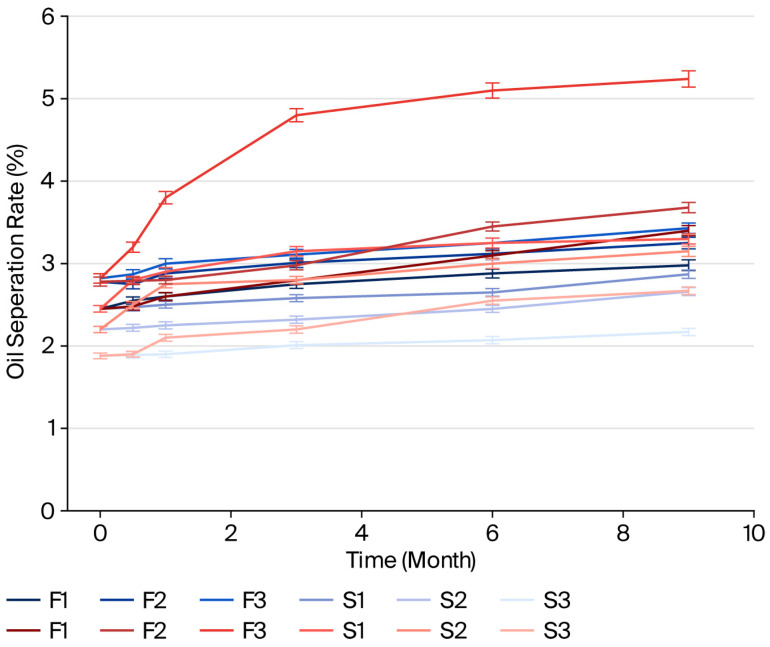
Oil separation rates of pistachio spreads (milk fat-based and sugar-based) at 4 °C (blue lines) and 25 °C (red lines) (the length of the error bars represents the standard error of the mean).

**Figure 3 foods-14-04002-f003:**
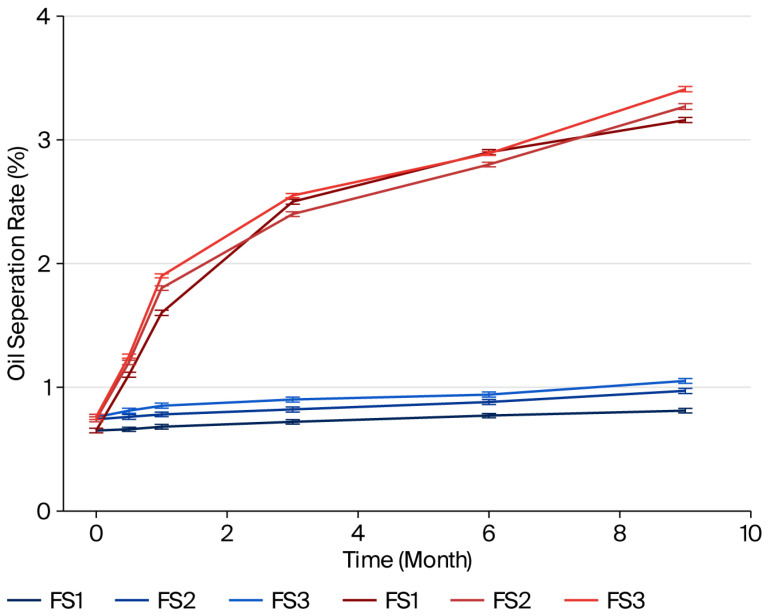
Oil separation rates of pistachio spreads (suga and milk fat-based) at 4 °C (blue lines) and 25 °C (red lines) (the length of the error bars represents the standard error of the mean).

**Figure 4 foods-14-04002-f004:**
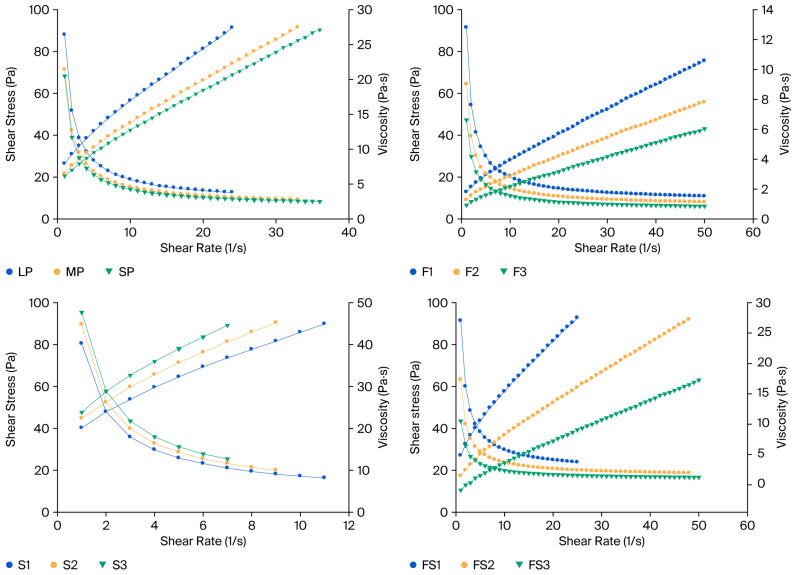
Changes in viscosity and shear stress with shear rate at 45 °C in pistachio paste and spread.

**Figure 5 foods-14-04002-f005:**
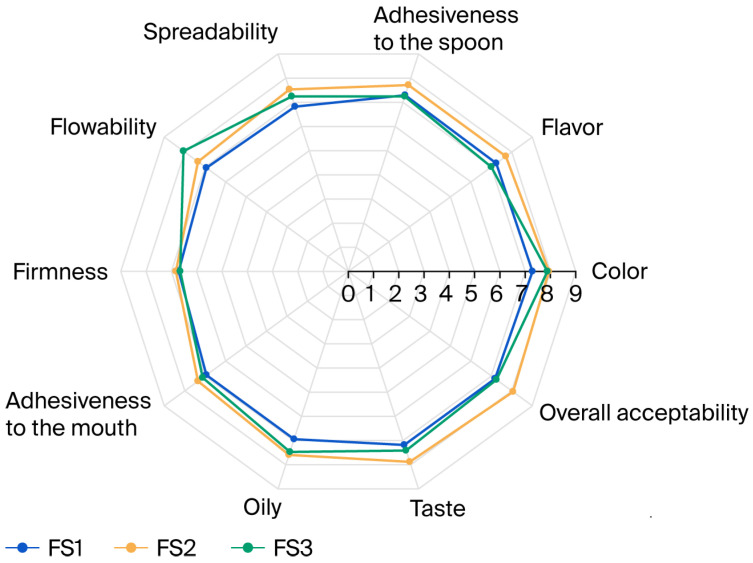
Sensory results from analysis of pistachio spreads. Standard errors of means for each attribute are as follows: color (0.17), flavor (0.12), adhesiveness to the spoon (0.19), spreadability (0.19), flowability (0.17), firmness (0.18), adhesiveness to the mouth (0.23), oily (0.12), and taste (0.16).

**Table 1 foods-14-04002-t001:** Particle size of pistachio paste as a function of cumulative particle volume (10%, 50%, and 90%) *.

Sample	D_10_ (µm)	D_50_ (µm)	D_90_ (µm)
LP	5.72 ± 0.06 ^c^	16.71 ± 0.12 ^c^	434.80 ± 9.30 ^c^
MP	2.02 ± 0.04 ^b^	11.89 ± 0.08 ^b^	395.40 ± 7.90 ^b^
SP	1.88 ± 0.03 ^a^	7.18 ± 0.07 ^a^	149.78 ± 4.60 ^a^

* All values are mean ± standard deviation of three replicates. Different superscript letters within the same column indicate statistical differences in the sample (*p* < 0.05).

**Table 2 foods-14-04002-t002:** Color values of pistachio pastes and spreads *.

Sample	L*	a*	b*
LP	44.34 ± 0.02 ^a^	−3.82 ± 0.05 ^a^	45.86 ± 0.05 ^g^
MP	44.41 ± 0.02 ^b^	−3.77 ± 0.03 ^ab^	44.91 ± 0.06 ^e^
SP	44.42 ± 0.03 ^b^	−3.64 ± 0.04 ^cde^	44.55 ± 0.05 ^c^
F1	44.76 ± 0.02 ^c^	−3.53 ± 0.07 ^f^	44.21 ± 0.06 ^a^
F2	44.91 ± 0.04 ^d^	−3.56 ± 0.08 ^ef^	44.37 ± 0.11 ^b^
F3	45.25 ± 0.03 ^e^	−3.72 ± 0.06 ^bc^	44.77 ± 0.09 ^d^
S1	46.79 ± 0.02 ^f^	−3.57 ± 0.03 ^ef^	44.19 ± 0.12 ^a^
S2	47.23 ± 0.03 ^g^	−3.66 ± 0.06 ^cd^	44.28 ± 0.09 ^ab^
S3	47.40 ± 0.03 ^h^	3.42 ± 0.04 ^g^	44.72 ± 0.08 ^d^
FS1	48.11 ± 0.03 ^j^	−3.61 ± 0.02 ^def^	44.73 ± 0.03 ^d^
FS2	48.28 ± 0.03 ^k^	−3.56 ± 0.03 ^ef^	44.64 ± 0.05 ^cd^
FS3	48.02 ± 0.04 ^i^	−3.59 ± 0.02 ^def^	45.10 ± 0.06 ^f^

* All values are mean ± standard deviation of three replicates. Different superscript letters within the same column indicate statistical differences in the sample (*p* < 0.05).

**Table 3 foods-14-04002-t003:** Regressed values of the Herschel–Bulkley model used to describe the flow curves of pistachio paste *.

Sample	Temperature (°C)	τ_0_ (Pa)	K	*n*
LP	20	19.80 ± 0.93 ^a^	15.70 ± 0.70 ^e^	0.770 ± 0.02 ^a^
	25	20.64 ± 0.56 ^a^	11.28 ± 0.66 ^d^	0.809 ± 0.06 ^a^
	30	20.28 ± 0.26 ^a^	9.28 ± 0.10 ^c^	0.803 ± 0.01 ^a^
	35	20.40 ± 0.11 ^a^	7.42 ± 0.10 ^b^	0.812 ± 0.01 ^a^
	40	20.52 ± 0.65 ^a^	6.50 ± 0.27 ^a^	0.797 ± 0.01 ^a^
	45	21.03 ± 0.57 ^a^	5.38 ± 0.21 ^a^	0.783 ± 0.01 ^a^
MP	20	19.64 ± 0.56 ^c^	11.47 ± 0.20 ^f^	0.823 ± 0.01 ^ab^
	25	18.07 ± 0.83 ^ab^	8.91 ± 0.29 ^e^	0.825 ± 0.01 ^ab^
	30	17.35 ± 0.40 ^a^	6.95 ± 0.10 ^d^	0.835 ± 0.01 ^b^
	35	17.26 ± 0.41 ^a^	5.65 ± 0.10 ^c^	0.835 ± 0.01 ^b^
	40	17.39 ± 0.10 ^a^	4.80 ± 0.10 ^b^	0.828 ± 0.01 ^ab^
	45	18.37 ± 0.29 ^b^	4.14 ± 0.15 ^a^	0.820 ± 0.01 ^a^
SP	20	19.62 ± 0.71 ^b^	9.76 ± 0.64 ^f^	0.848 ± 0.04 ^a^
	25	17.13 ± 0.82 ^a^	7.94 ± 0.25 ^e^	0.837 ± 0.01 ^a^
	30	16.56 ± 0.85 ^a^	6.18 ± 0.13 ^d^	0.847 ± 0.01 ^a^
	35	16.25 ± 0.68 ^a^	5.10 ± 0.14 ^c^	0.844 ± 0.01 ^a^
	40	16.48 ± 0.23 ^a^	4.20 ± 0.10 ^b^	0.846 ± 0.01 ^a^
	45	17.48 ± 0.12 ^a^	3.53 ± 0.01 ^a^	0.844 ± 0.01 ^a^

* All values are mean ± standard deviation of three replicates. Different superscript letters within the same column indicate statistical differences depending on temperature for each sample (*p* < 0.05).

**Table 4 foods-14-04002-t004:** Flow behavior parameters for pistachio spread (milk fat-based) according to the Herschel–Bulkley model *.

Sample	Temperature (°C)	τ_0_ (Pa)	K	*n*
F1	20	12.71 ± 0.69 ^b^	6.05 ± 0.01 ^f^	0.868 ± 0.001 ^b^
	25	12.57 ± 0.20 ^b^	4.46 ± 0.07 ^e^	0.884 ± 0.004 ^d^
	30	11.45 ± 0.26 ^a^	3.65 ± 0.06 ^d^	0.880 ± 0.004 ^cd^
	35	11.25 ± 0.21 ^a^	2.98 ± 0.06 ^c^	0.879 ± 0.003 ^c^
	40	11.47 ± 0.10 ^a^	2.45 ± 0.02 ^b^	0.878 ± 0.002 ^c^
	45	11.52 ± 0.20 ^a^	2.22 ± 0.02 ^a^	0.858 ± 0.001 ^a^
F2	20	9.73 ± 0.05 ^c^	4.76 ± 0.05 ^f^	0.871 ± 0.003 ^b^
	25	9.06 ± 0.20 ^b^	3.19 ± 0.01 ^e^	0.893 ± 0.001 ^c^
	30	8.75 ± 0.07 ^b^	2.46 ± 0.03 ^d^	0.901 ± 0.002 ^d^
	35	8.92 ± 0.40 ^b^	2.00 ± 0.06 ^c^	0.900 ± 0.006 ^d^
	40	8.19 ± 0.03 ^a^	1.86 ± 0.01 ^b^	0.872 ± 0.001 ^b^
	45	8.24 ± 0.05 ^a^	1.59 ± 0.09 ^a^	0.852 ± 0.005 ^a^
F3	20	10.51 ± 0.16 ^d^	6.88 ± 0.22 ^e^	0.801 ± 0.007 ^a^
	25	8.10 ± 0.14 ^c^	2.73 ± 0.11 ^d^	0.897 ± 0.008 ^e^
	30	6.41 ± 0.02 ^b^	1.94 ± 0.01 ^c^	0.896 ± 0.002 ^e^
	35	6.11 ± 0.07 ^a^	1.69 ± 0.02 ^b^	0.880 ± 0.002 ^d^
	40	6.11 ± 0.02 ^a^	1.47 ± 0.01 ^a^	0.867 ± 0.002 ^c^
	45	6.05 ± 0.02 ^a^	1.33 ± 0.02 ^a^	0.849 ± 0.002 ^b^

* All values are mean ± standard deviation of three replicates. Different superscript letters within the same column indicate statistical differences depending on temperature for each sample (*p* < 0.05).

**Table 5 foods-14-04002-t005:** Flow behavior parameters for pistachio spread (sugar-based), according to the Herschel–Bulkley model *.

Sample	Temperature (°C)	τ_0_ (Pa)	K	*n*
S1	20	nd	nd	nd
	25	nd	nd	nd
	30	26.94 ± 0.86 ^bc^	17.28 ± 0.73 ^d^	0.710 ± 0.015 ^d^
	35	25.22 ± 0.05 ^a^	15.43 ± 0.04 ^c^	0.693 ± 0.001 ^c^
	40	26.04 ± 0.33 ^ab^	13.90 ± 0.25 ^b^	0.674 ± 0.003 ^b^
	45	27.46 ± 0.52 ^c^	13.03 ± 0.27 ^a^	0.642 ± 0.010 ^a^
S2	20	nd	nd	nd
	25	nd	nd	nd
	30	28.48 ± 0.20 ^a^	20.01 ± 0.50 ^d^	0.689 ± 0.014 ^c^
	35	27.60 ± 0.39 ^a^	17.54 ± 0.21 ^c^	0.675 ± 0.010 ^bc^
	40	27.70 ± 0.30 ^a^	15.88 ± 0.29 ^b^	0.662 ± 0.010 ^ab^
	45	29.80 ± 0.84 ^b^	14.62 ± 0.52 ^a^	0.648 ± 0.011 ^a^
S3	20	nd	nd	nd
	25	nd	nd	nd
	30	nd	nd	nd
	35	24.20 ± 0.33 ^a^	25.16 ± 0.62 ^c^	0.581 ± 0.010 ^a^
	40	24.96 ± 0.37 ^a^	22.58 ± 0.35 ^b^	0.572 ± 0.010 ^a^
	45	27.82 ± 0.82 ^b^	19.96 ± 0.76 ^a^	0.581 ± 0.019 ^a^

* All values are mean ± standard deviation of three replicates. Different superscript letters within the same column indicate statistical differences depending on temperature for each sample (*p* < 0.05). nd: not determined.

**Table 6 foods-14-04002-t006:** Flow behavior parameters for pistachio spread (sugar- and milk fat-based) according to the Herschel–Bulkley model *.

Sample	Temperature (°C)	τ_0_ (Pa)	K	*n*
FS1	20	16.48 ± 0.03 ^a^	19.01 ± 0.21 ^f^	0.692 ± 0.004 ^a^
	25	18.98 ± 0.02 ^b^	12.96 ± 0.20 ^e^	0.751 ± 0.005 ^cd^
	30	19.17 ± 0.38 ^bc^	10.18 ± 0.21 ^d^	0.759 ± 0.007 ^d^
	35	19.68 ± 0.16 ^cd^	8.37 ± 0.17 ^c^	0.763 ± 0.006 ^d^
	40	19.82 ± 0.11 ^d^	7.61 ± 0.07 ^b^	0.740 ± 0.002 ^c^
	45	20.47 ± 0.54 ^e^	7.20 ± 0.32 ^a^	0.715 ± 0.011 ^b^
FS2	20	14.45 ± 0.39 ^b^	9.12 ± 0.46 ^e^	0.811 ± 0.012 ^a^
	25	14.20 ± 0.10 ^ab^	7.06 ± 0.05 ^e^	0.823 ± 0.002 ^a^
	30	13.91 ± 0.13 ^a^	5.06 ± 0.10 ^d^	0.838 ± 0.009 ^b^
	35	14.60 ± 0.26 ^bc^	4.02 ± 0.05 ^c^	0.848 ± 0.003 ^b^
	40	14.90 ± 0.23 ^cd^	3.39 ± 0.06 ^b^	0.845 ± 0.004 ^b^
	45	15.07 ± 0.20 ^d^	3.16 ± 0.04 ^a^	0.823 ± 0.003 ^a^
FS3	20	16.78 ± 0.41 ^b^	15.22 ± 0.19 ^f^	0.716 ± 0.007 ^a^
	25	13.40 ± 0.14 ^b^	6.16 ± 0.05 ^e^	0.814 ± 0.002 ^b^
	30	9.52 ± 0.20 ^a^	2.97 ± 0.04 ^d^	0.879 ± 0.020 ^e^
	35	9.39 ± 0.02 ^a^	2.48 ± 0.02 ^c^	0.872 ± 0.002 ^e^
	40	9.30 ± 0.02 ^a^	2.19 ± 0.01 ^b^	0.858 ± 0.002 ^d^
	45	9.38 ± 0.02 ^a^	2.03 ± 0.04 ^a^	0.836 ± 0.005 ^c^

* All values are mean ± standard deviation of three replicates. Different superscript letters within the same column indicate statistical differences depending on temperature for each sample (*p* < 0.05).

**Table 7 foods-14-04002-t007:** Arrhenius parameters of pistachio paste and spreads.

Sample	*Ea*	k_0_	r2
LP	32.1	2.8 × 10^−5^	0.9866
MP	31.8	2.4 × 10^−5^	0.9926
SP	31.8	2.1 × 10^−5^	0.9986
F1	27.5	6.4 × 10^−5^	0.9860
F2	22.2	3.6 × 10^−4^	0.9714
F3	20.4	5.9 × 10^−4^	0.9963
S1	15.3	4.0 × 10^−2^	0.9891
S2	18.4	2.6 × 10^−2^	0.9916
S3	18.9	1.6 × 10^−2^	0.9978
FS1	18.2	7.0 × 10^−3^	0.9336
FS2	25.4	2.0 × 10^−4^	0.9569
FS3	20.3	9.0 × 10^−4^	0.9720

**Table 8 foods-14-04002-t008:** Textural properties of pistachio paste and spread *.

Samples	Firmness (N)	Spreadability (N·s)	Adhesiveness (N·s)
LP	2.69 ± 0.04 ^f^	1.589 ± 0.021 ^f^	−0.667 ± 0.012 ^g^
MP	2.68 ± 0.04 ^f^	1.568 ± 0.019 ^f^	−0.631 ± 0.011 ^f^
SP	2.36 ± 0.03 ^e^	1.432 ± 0.018 ^e^	−0.538 ± 0.009 ^e^
F1	1.79 ± 0.02 ^d^	1.064 ± 0.017 ^d^	−0.401 ± 0.008 ^d^
F2	1.35 ± 0.02 ^b^	0.830 ± 0.015 ^b^	−0.296 ± 0.004 ^b^
F3	1.02 ± 0.02 ^a^	0.630 ± 0.012 ^a^	−0.224 ± 0.004 ^a^
S1	5.05 ± 0.06 ^h^	2.995 ± 0.032 ^h^	−1.272 ± 0.012 ^j^
S2	5.10 ± 0.06 ^h^	3.014 ± 0.041 ^h^	−1.212 ± 0.011 ^i^
S3	5.33 ± 0.05 ^i^	3.192 ± 0.040 ^i^	−1.300 ± 0.014 ^k^
FS1	3.66 ± 0.04 ^g^	2.143 ± 0.028 ^g^	−0.796 ± 0.010 ^h^
FS2	2.39 ± 0.03 ^e^	1.426 ± 0.017 ^e^	−0.544 ± 0.008 ^e^
FS3	1.65 ± 0.03 ^c^	0.970 ± 0.016 ^c^	−0.352 ± 0.004 ^c^

* All values are the mean ± standard deviation of three replicates. Different superscript letters within the same column indicate statistical differences for samples (*p* < 0.05).

## Data Availability

The original contributions presented in the study are included in the article/Appendix A. Further inquiries can be directed to the corresponding author.

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
