# Peer review of "Impact of Formulation on the Rheological, Textural, and Sensory Properties of Pistachio Spread"

_foods, 2025, doi:10.3390/foods14234002_

Round 1
Reviewer 1 Report (Previous Reviewer 1)
Comments and Suggestions for Authors
From the standpoint of overall structure and scholarly quality, this manuscript demonstrates a rigorous experimental design and strong data support, reflecting solid experimental skills and professionalism in food engineering. By using Boz pistachio for a multifactorial study, the authors address a gap concerning the interactive effects of particle size–fat–sucrose–temperature on rheological behavior, which carries clear industrial relevance. I recommend that the revision further strengthen results analysis, theoretical interpretation, and articulation of innovation. Major revision is recommended.
1.Introduction need strengthen significance and knowledge gap. The current introduction clearly presents the background and economic value of Boz Antep pistachio, but the research gap is only briefly mentioned at the end. Explicitly state how this work differs from prior literature, for example, “previous studies focused on single variables, whereas this study systematically examines the interactions among particle size, milk fat, and sucrose.” Also emphasize the contribution to clean-label product development.
2. Update the literature base. Citations cluster around ~2015; important work from the last three years is missing.
3. Make the hypotheses more specific. The manuscript posits “finer particle size, higher milk fat, and higher sucrose, to change in flow behavior,” but does not specify expected quantitative relationships.
4. Justify the experimental design. Explain why 20-45oC was chosen as the test temperature range and why the selected fat and sucrose levels were used. Add details on randomization and the number of replicates to improve reproducibility and statistical credibility.
5. Deepen mechanistic interpretation. Use fat-crystal structure theory or particle-dispersion models to explain how milk fat and sucrose affect yield stress. Concepts from emulsion kinetics or glass transition temperature can elevate the level of analysis.
6. Improve data visualization. Several result tables are dense and lack visual guidance.
7. Highlight innovation in the abstract and conclusion. Make the study’s novelty explicit.
8. Add practice-oriented guidance. Provide actionable design recommendations for industrial application.
9. Tighten logic and flow. The prose is clear but some logical transitions are abrupt.
10. Expand micro-scale mechanisms of rheology. Provide a more in-depth discussion of the microstructural mechanisms underlying the observed rheological behavior.
Author Response
Dear Reviewer
We greatly appreciate your thorough assessment and insightful suggestions, which significantly enhanced the quality of our manuscript. We have carefully evaluated each remark and revised the content accordingly.
All requested changes have been incorporated into the updated version. The manuscript underwent professional editing for English language quality (MDPI Author Services, ID: english-103123), and all tables and figures were thoroughly revised using the MDPI Table/Figure Editing Service to insure accuracy and consistency.
We provide a comprehensive and detailed response to each reviewer’s comment. Modifications in the modified manuscript are indicated in yellow and green. Additionally, after final editing, the manuscript was finalized.
Please note that the explanations and responses originally submitted to the reviewers were composed prior to the professional English editing procedure. Consequently, specific words, sentences, and syntactic structures may manifest as somewhat altered in the final iteration of the manuscript owing to linguistic enhancement. Furthermore, all tables and figures have been refined using MDPI’s expert Table/Figure Editing Service, and their final edited versions are incorporated in the resubmitted article.
The editorial adjustments have not changed the scientific content or data interpretation but have enhanced clarity, consistency, and visual display. We value the helpful criticism provided by the reviewers and the editor. We believe that the adjustments have significantly enhanced the scientific clarity, precision, and presentation of our study.
Sincerely Yours,
1- Introduction need strengthen significance and knowledge gap. The current introduction clearly presents the background and economic value of Boz Antep pistachio, but the research gap is only briefly mentioned at the end. Explicitly state how this work differs from prior literature, for example, “previous studies focused on single variables, whereas this study systematically examines the interactions among particle size, milk fat, and sucrose.” Also emphasize the contribution to clean-label product development.
Thank you very much for your valuable suggestion. The Introduction section has been revised in line with your suggestions. In addition, the following explanatory paragraph has been added as per your suggestion.
Previous studies on pistachio-based formulations have primarily investigated single variables such as fat level, particle size, or sugar content in isolation. In contrast, the present study systematically examines the combined effects of particle size, milk fat, and sucrose, allowing for a deeper understanding of their interactive influence on rheology and texture. This integrated approach provides unique insight into clean-label pistachio spread development, where formulation must balance texture, stability, and sensory quality without additives.
- Update the literature base. Citations cluster around ~2015; important work from the last three years is missing.
Thank you very much for your suggestions. The current literature has been re-examined and the following references, which will contribute to our study, have been added. Citations have been made in the appropriate places within the text.
“Recent studies increasingly focus on clean-label nut spreads and fat-replacement strategies, including oleogel-based systems and sugar alternatives, reflecting growing industrial interest in healthier spread formulations [Malvano et al., 2024; Berk et al., 2024; Prakansamut et al., 2024; Tonetto et al. (2025). Additionally, advances in lipid crystallization and emulsion structuring technologies have improved understanding of fat–solid interactions in nut-based matrices [da Silva et al., 2024]. However, limited research specifically examines pistachio-based systems, particularly regarding the combined roles of particle size, milk fat, and sucrose on rheology and textural attributes, highlighting the need for multidimensional investigations such as the present study.”
“The results of the present study are consistent with recent investigations highlighting the role of lipid network structuring and sugar–fat interactions in determining the spreadability, stability, and viscoelastic properties of nut-based spreads [Marra et al., 2023; Liu et al., 2024]. In agreement with these outcomes, research on hazelnut and chocolate spreads has demonstrated that fat composition and microstructural network development significantly influence firmness, flow behavior, and melting characteristics [Berk et al., 2024; Prakansamut et al., 2024; MaÅ‚kowska, M.,2021]. Moreover, recent work emphasizes the importance of fat crystallization mechanisms and the distinct functional behavior of polyunsaturated versus saturated lipids in shaping textural stability and consumer perception in spreadable systems [Tonetto et al., 2025; da Silva & Martini, 2024]. Collectively, these findings underscore the relevance of the present research and reflect the growing interest in clean-label, naturally structured nut-based spreads in both scientific and industrial contexts.
“In line with recent clean-label nut spread formulation trends, the natural fat-sugar balance and particle structure were important factors in the present research's sensory acceptance [Malvano et al., 2024; Berk et al., 2024].”
-Malvano, F., Muccio, E., Galgano, F., Marra, F., & Albanese, D. (2024). Design of a high protein, no added sugar pistachio spread using oleogel as fat replacer. LWT, 198, 115993. https://doi.org/10.1016/j.lwt.2024.115993
- Liu, L., Gao, Z., Chen, G., Yao, J., Zhang, X., Qiu, X., & Liu, L. (2024). A comprehensive review: Impact of oleogel application on food texture and sensory properties. Food Science & Nutrition, 12(6), 3849–3862. https://doi.org/10.1002/fsn3.4110
- Berk, B., Cosar, S., Mazı, B. G., & Oztop, M. H. (2024). Textural, rheological, melting properties, particle size distribution and NMR of cocoa-hazelnut spread with inulin/stevia. Journal of Texture Studies, 55(2). https://doi.org/10.1111/jtxs.12834
-Prakansamut, N., Adulpadungsak, K., Sonwai, S., Aryusuk, K., & Lilitchan, S. (2024). Application of functional oil blend-based oleogels as novel structured oil alternatives in chocolate spread. LWT, 203, 116322. https://doi.org/10.1016/j.lwt.2024.116322
- da Silva, T. L. T., & Martini, S. (2024). Recent Advances in Lipid Crystallization in the Food Industry. Annual Review of Food Science and Technology, 15(1), 355–379. https://doi.org/10.1146/annurev-food-072023-034403
-Tonetto, M.L., Teixeira, G.L., Kechinski, C.P. et al. Turning nuts and peanuts into functional spreads with bioactive compounds as an opportunity for innovation. Discov Food 5, 131 (2025). https://doi.org/10.1007/s44187-025-00414-6
MaÅ‚kowska, M., Staniewski, B., & Ziajka, J. (2021). Analyses of milk fat crystallization and milk fat fractions. International Journal of Food Properties, 24(1), 325–336. https://doi.org/10.1080/10942912.2021.1878217
- Make the hypotheses more specific. The manuscript posits “finer particle size, higher milk fat, and higher sucrose, to change in flow behavior,” but does not specify expected quantitative relationships.
Thank you very much for your suggestions and evaluation. The introduction section has been revised accordingly. A hypothesis sentence appropriate to the purpose of the study has been added as follows at the end of introduction.
“Therefore, this study aimed to systematically evaluate the effects of particle size, milk fat, and sucrose on the rheological and textural characteristics of pistachio spreads across different temperatures, based on the hypothesis that finer particle size and higher milk fat levels would reduce yield stress and consistency (enhancing spreadability), whereas higher sucrose concentration would increase structural strength and decrease flowability.”
- Justify the experimental design. Explain why 20-45°C was chosen as the test temperature range and why the selected fat and sucrose levels were used. Add details on randomization and the number of replicates to improve reproducibility and statistical credibility.
The current study was guided by sensory evaluations conducted at various stages of the formulation process. Initial trials involved a pistachio paste with a large particle size, and sensory tests were conducted with the panelists. Panelists reported that the product had a very dense texture that was sticky and left a residual sensation in the mouth. To make the texture more palatable and spreadable (and more compact and sustainable), it was decided to use sugar and fat. It was determined that milk fat was the most suitable fat to use for maintaining a compact structure in a spreadable pistachio butter. Milk fat was used at a maximum of 10% (w/w) and a minimum of 4% (w/w). This decision was based on the observation that adding 10% (w/w) or more of milk fat resulted in a very dominant, strong fat taste in the product.
To individually evaluate the effect of the components in the product structure, trials were conducted with different ratios of milk fat and sugar. According to the sensory test results, the most preferred product, designated as FS2, consisted of 66% pistachio paste (small particle size), 27% sugar, and 7% milk fat content. All subsequent formulation selections were based on this preferred product structure. Panelists were subjected to tasting and product-specific tests.
The temperature range of 20 to 45°C was chosen to include the most important conditions for storing and using spreadable foods and also temperature fluctuations during summers. 20 to 25°C ranges shows the normal temperature for spreadable products to be used at and serves as a baseline for testing texture and rheology in normal use. 30-35 °C ranges; for determining structural stability and potential phase separation (like fat migration, melting and etc.) while >45 °C increased heat-induced instabilities (e.g., rapid softening/oil release).
Section 2.9 also states that the analyses were performed triplicate. “All experiments were conducted in triplicate, and the results are presented as mean values”
- Deepen mechanistic interpretation. Use fat-crystal structure theory or particle-dispersion models to explain how milk fat and sucrose affect yield stress. Concepts from emulsion kinetics or glass transition temperature can elevate the level of analysis.
Thank you very much for your comments. The following section has been added and revised in the relevant section of the results.
The temperature-dependent decline in the flow behavior index (n) indicated stronger shear-thinning behavior at lower temperatures (20–30 °C) and a gradual shift toward more Newtonian flow at higher temperatures (40–45 °C). The rheological behavior of the pistachio spreads can be explained by the combined effects of particle morphology, fat crystallization, and sucrose-mediated solid interactions. Finer particle size increased yield stress (τâ‚€) and consistency due to tighter particle packing and larger surface area, leading to a more continuous and cohesive structural network. This observation aligns with particle-dispersion theory, where smaller particles enhance interparticle friction and structural rigidity.
Milk fat primarily acted as a softening and lubricating phase. According to fat-crystal network theory, the proportion and morphology of solid fat govern the firmness and temperature sensitivity of fat-based systems. As noted by MaÅ‚kowska et al. (2021), milk fat crystallization involves several polymorphic forms (α, β′, β), whose transitions influence the mechanical stability of the lipid matrix. At lower temperatures, higher solid-fat content reinforces the network and increases τâ‚€, while at temperatures above 40 °C, crystal melting and polymorphic conversion weaken the structure, thereby reducing yield stress and viscosity.
Conversely, sucrose strengthened the solid framework by promoting crystalline bridging and particle adhesion, which increased τâ‚€ and decreased spreadability. Because sucrose remains crystalline within the tested temperature range (20–45 °C), it contributes rigidity without undergoing glass-transition softening.
Overall, these findings indicate that smaller particle size and higher sugar concentration reinforce the solid matrix, whereas higher milk fat levels promote flow by disrupting crystal–particle interactions. The textural and rheological properties of pistachio spreads are therefore determined by the balance between these opposing mechanisms.
- Improve data visualization. Several result tables are dense and lack visual guidance.
Figures and tables have been reorganized.
- Highlight innovation in the abstract and conclusion. Make the study’s novelty explicit.
Thank you very much for your explanatory suggestion, which will make the study more valuable. The summary has been organized as follows.
Abstract;
“This research represents the first systematic evaluation of the combined effects of particle size, milk fat, and sucrose on the rheological and textural behavior of Boz Antep pistachio spreads produced under clean-label conditions, offering new insights into the structural design of additive-free nut-based systems.
These findings highlight the interplay between ingredient composition and physical properties, providing valuable guidance for developing high-quality pistachio-based spreads.”
Conclusion
“This research provides the first integrated evaluation of multi-factor interactions in Boz Antep pistachio spreads under clean-label conditions, offering a mechanistic understanding that links formulation variables to rheological and textural performance. These insights contribute to the scientific foundation for developing naturally structured, additive-free nut spreads with improved consumer appeal and industrial applicability.”
- Add practice-oriented guidance. Provide actionable design recommendations for industrial application.
Thank you for this suggestion. The section on industrial applications addressing the results of the study has been added to the end of the results section.
“From an industrial perspective, these results can be used to develop guidelines for formulation and process design. For example, maintaining medium particle sizes and moderate milk fat levels (6–8%) can make the product easier to spread and prevent the oil from separating. On the other hand, adjusting the sucrose concentration (25–28%) can assist the product get an acceptable structural strength without being too sticky. The stated temperature range (20–45 °C) also provides a practical way to manage the flow of materials when combining, filling, and storing. The results provide actionable insights for creating standardized, clean-label pistachio spreads that exhibit consistent rheological properties and a texture acceptable to consumers.”
- Tighten logic and flow. The prose is clear but some logical transitions are abrupt.
We appreciate the reviewer's helpful criticism. The Introduction, Results and Discussion, and Conclusion portions of the manuscript have all undergone extensive revisions to enhance logical coherence and narrative flow. To improve the relationships among formulation factors, observed rheological trends, and associated sensory results, transitional sentences were incorporated. Additionally, in order to ensure a more seamless transition from background to hypothesis and from findings to implications, superfluous phrases were eliminated and paragraph sequencing was modified. These changes improve the text's readability and bring it closer to the experimental interpretation's logical framework.
- Expand micro-scale mechanisms of rheology. Provide a more in-depth discussion of the microstructural mechanisms underlying the observed rheological behavior.
We appreciate the reviewer's insightful recommendation. To provide clarity on the microstructural mechanisms behind the observed flow behavior, a new explanatory paragraph has been included to the Rheology section of the updated text. The synergistic impacts of fat-crystal polymorphism, sucrose-mediated solid bridging, and particle morphology on yield stress and consistency are covered in this additional section. In addition to the explanations added to the results and discussion section, two of the sections added below are shown as examples.
“This behavior reflects the dual role of solid milk-fat crystals: at low temperatures, crystal structures act as rigid fillers that increase network strength and limit particle mobility, while at higher temperatures, melted fat contributes to lubrication and reduces internal friction, facilitating flow and spreadability. Such polymorphism-driven transitions are characteristic of lipid-rich colloidal systems.”
“This increases interparticle cohesion and reduces molecular mobility, thus providing higher hardness and energy requirements during deformation. This partly explains the formation of a regular sucrose matrix, higher viscosity, and lower spreadability observed in high sucrose formulations.”

Reviewer 2 Report (Previous Reviewer 3)
Comments and Suggestions for Authors
I am glad that you revised the manuscript in accordance with the requirements.
Author Response
We express our heartfelt gratitude for your thoughtful message and for recognizing that the article has been amended to meet the journal's specifications. We sincerely value the time and effort you dedicated to evaluating our work and offering constructive suggestions that enhanced the scientific quality and clarity of the publication.
In this final revision, all outstanding editorial and formatting modifications have been carefully executed. The manuscript has received professional English editing and table/figure formatting services from MDPI Author Services to guarantee linguistic precision and visual uniformity.
We have thoroughly verified every reviewer recommendations to ensure that each comment has been adequately handled and included into the final content. We assert that these extensive edits have significantly improved the manuscript's readability and scientific rigor.
We like to express our deepest gratitude for your invaluable help during the evaluation process. Please find attached the file containing the detailed changes made to the manuscript and its final revised version.
Sincerely Yours,

Reviewer 3 Report (Previous Reviewer 2)
Comments and Suggestions for Authors
The authors of the manuscript propose the hypothesis that formulating a pistachio spread may be of interest to the food industry in order to provide a product of standardized quality. The objective of the study was to evaluate the impact of the formulation on the rheological, textural, and sensory properties of the pistachio spread. An experimental design was developed to achieve this objective. However, the study has two weaknesses that the authors should address.
The first concern is that the sensory analysis test is not a Quantitative Descriptive Analysis (QDA), but rather a Hedonic Test (HT). While QDA requires an unstructured scale with only two anchor words, HT involves a structured Likert-type scale (a 9-point scale). Additionally, the statistical analysis of both tests is contingent on the type of scale utilized (continuous vs. discrete).
The second concern is that the authors mention a correlation between physical parameters and sensory attributes. However, they do not provide a detailed explanation of the statistical method applied to establish the association.
Another issue that the authors should improve is the discussion of rheological and textural behaviors as a function of the lipid profile of spreads. Pistachio fat is high in polyunsaturated fatty acids, and dairy fat is rich in saturated fatty acids. Although these profiles have not been assessed, the authors could suggest an explanation.
To facilitate the enhancement of the manuscript, a detailed report is provided to the authors.

Author Response
Dear Reviewer
We greatly appreciate your thorough assessment and insightful suggestions, which significantly enhanced the quality of our manuscript. We have carefully evaluated each remark and revised the content accordingly.
All requested changes have been incorporated into the updated version. The manuscript underwent professional editing for English language quality (MDPI Author Services, ID: english-103123), and all tables and figures were thoroughly revised using the MDPI Table/Figure Editing Service to insure accuracy and consistency.
We provide a comprehensive and detailed response to each reviewer’s comment. Modifications in the modified manuscript are indicated in yellow and green. Additionally, after final editing, the manuscript was finalized.
Please note that the explanations and responses originally submitted to the reviewers were composed prior to the professional English editing procedure. Consequently, specific words, sentences, and syntactic structures may manifest as somewhat altered in the final iteration of the manuscript owing to linguistic enhancement. Furthermore, all tables and figures have been refined using MDPI’s expert Table/Figure Editing Service, and their final edited versions are incorporated in the resubmitted article.
The editorial adjustments have not changed the scientific content or data interpretation but have enhanced clarity, consistency, and visual display. We value the helpful criticism provided by the reviewers and the editor. We believe that the adjustments have significantly enhanced the scientific clarity, precision, and presentation of our study.
Sincerely Yours,
Page 1-
Number 1-Sensory properties were also evaluated. Rewrite the title. e.g.
Impact of formulation on the rheological, textural, and sensory properties of pistachio spread
The title has been revised
Impact of Formulation on the Rheological, Textural, and Sensory Properties of Pistachio Spread
2- Note that the information in lines 17-18 is already indicated in line 14. Delete it and complete line 14, providing the percentages of added fat and sugar.
The text has been edited as suggested.
The present study investigates the effects of particle size, milk fat, and sucrose content on
the rheological and textural properties of pistachio spread formulations. Three pistachio
pastes were prepared using a mill with different particle sizes: LP (large), MP (medium),
and SP (small). Pistachio spreads were subsequently formulated at three milk-fat levels (4%, 7%, and 14%) and three sucrose levels (20%, 27%, and 30%). In the combined formulations, 66% pistachio paste was fixed, and three combinations were prepared (FS1: 4% milk fat, 30% sucrose; FS2: 7% milk fat, 27% sucrose; FS3: 14% milk fat, 20% sucrose) to assess the individual and interactive effects of these components.
Number: 3
The method for establishing the correlation between textural and rheological properties and sensory properties has not been indicated.
We appreciate your comment. We compared the results of sensory acceptance with objective data rather than doing a statistical correlation study. Instead of suggesting statistical association, the abstract's language has been changed to make it clear that both the instrumental and sensory studies demonstrated consistent trends.
Sensory data indicated patterns aligned with objective textural and rheological measurements.
Number: 4
Author: Include “sensory attribute”
“sensory attribute” has been added to the keywords
Number: 5
Number: To ensure that the paper is searchable in databases, use different words in the title and keywords.
“Nutspreads” has been added to the keywords
Number: 1
Summarize this information and highlight the interest in developing a pistachio paste and spread. The authors should explain the differences between the two types of products. The authors should also remark that studies conducted with both types of products are still insufficient.
Thank you very much for your valuable comment. This section has been revised to clearly distinguish between pistachio paste and pistachio cream, emphasizing the industrial importance of developing these products and highlighting the current limited number of studies on both. A summary has been added to strengthen the rationale and innovation of the current study.
“Rheology, the study of deformation and flow, is essential for understanding the physicochemical and structural behavior of foods [9]. Pistachio products exhibit complex viscoelastic properties that affect stability, spreadability, and sensory perception. Previous work has modeled the behavior of pistachio paste using Herschel–Bulkley flow parameters, highlighting its yield stress and shear-thinning nature [10–11]. Texture, a multisensory characteristic shaped by mechanical and structural properties, is essential for consumer acceptance [12–13]. Instrumental texture analysis and sensory testing are commonly used to evaluate firmness, adhesiveness, and spreadability in pistachio-based systems [14].
Pistachio paste consists solely of ground pistachio kernels, whereas pistachio spread includes additional components, such as fat and sugar, to enhance its structure and flavor. Particle size distribution, fat–solid interactions, and formulation composition strongly influence rheology and texture in nut-based matrices [15–21], which are typically considered semi-solid, oil-continuous systems. Stabilization strategies using emulsifiers and oleogelators (e.g., monoglycerides, lecithin, beeswax, and rice bran wax) have been explored to reduce oil separation and improve storage stability [15, 22–23]. Adjusting fat content, moisture level, and particle size has also been shown to affect yield stress, hardness, and spreadability [19].
Despite this knowledge, research specifically focused on pistachio paste and pistachio spread remains limited compared with other nut systems (e.g., peanut, hazelnut), indicating a need for more systematic studies to support the development of standardized, and high-quality pistachio products.”
Number: 2
Move to the end of the introduction section
It was added to the end of the introduction.
Page: 3
Number: 1
This paragraph is a summary of the study to be conducted. Consider whether it is more appropriate for the discussion of the results and conclusions.
Thank you for your comment. We appreciate your suggestion regarding the placement of this paragraph. Previous referee comments requested that similar details be explicitly added to the Introduction section to clarify the concept of clean label formulation and the uniqueness of using only pistachio paste, milk fat, and sucrose. Following your suggestion, the paragraph has been revised and simplified for the Introduction section to avoid result-oriented statements and maintain an appropriate academic structure. Interpretive statements have been removed, and the content now focuses solely on clarifying the formulation strategy and scope of the study.
“In this study, pistachio spreads were developed using pistachio pastes, formulated with only milk fat and sucrose, without the addition of emulsifiers, stabilizers, or additives. This clean-label approach enables the investigation of the intrinsic effects of formulation variables on the rheological, textural, and sensory properties of pistachio-based spreads. Given the growing consumer and industrial interest in natural nut-based products, examining the behavior of additive-free pistachio formulations is crucial for supporting the development of high-quality, minimally processed pistachio spreads.”
Number: 2
Rewrite in a simple way as: "Anhydrous milk fat and powdered sugar were added to the SP pistachio paste to elaborate three formulations of the pistachio spread (PS) (Table 1).
(Instead of, Anhydrous cow’s milk fat, icing sugar were added to the pistachio paste (SP) according to the formulations as shown in Table 1.)
The sentence was corrected as indicated.
Page: 4
Number: 1 Move to the supplementary material section and rename as Table S1.
Moved to the supplementary material section and renamed as Table S1.
Number: 2 Rewrite as: "Ingredients (%, w/w) used in the preparation of SP pistachio spread formulations."
The sentence was corrected as indicated.
Number: 3 Move to the header of the table.
Moved to the header of the table.
Nuber: 4 Delete
The table has been corrected as specified and is now included as supplementary material.
Table S1. Ingredients (%, w/w) used in the preparation of SP pistachio spread formulations."
|
Formulation |
Pistachio paste |
Milk fat |
Icing sugar |
|
F1 |
96 |
4 |
- |
|
F2 |
93 |
7 |
- |
|
F3 |
90 |
10 |
- |
|
S1 |
76 |
- |
24 |
|
S2 |
73 |
- |
27 |
|
S3 |
70 |
- |
30 |
|
FS1 |
66 |
4 |
30 |
|
FS2 |
66 |
7 |
27 |
|
FS3 |
66 |
10 |
24 |
Number: 5 Replace with "samples of pistachio paste and spread"
The sentence was corrected as indicated.
Separate centrifuge tubes were filled with roughly 15 g of each samples of pistachio paste and spread and were heated to 80°C in a water bath for 30 min.
Page: 5
Number: 1 Rewrite as equations (3) and (4)
The sentence was corrected as indicated in the manuscript.
Page 6.
Number: 1
According to the description provided in this section, the method used does not meet the requirements of what is known as QDA. I advise the authors to consult the following chapter:
Stone, H., Bleibaum, R. N., & Thomas, H. A. (2021). Descriptive analysis. In H. Stone, R. N. Bleibaum, & H. A. Thomas (Eds.), Sensory Evaluation Practices (5th
ed., pp. 235–295). Academic Press. https://doi.org/10.1016/B978-0-12-815334-5.00001-X
We appreciate your comment and read the chapter.
We agree that the sensory analysis conducted in this study corresponds to a 9-point hedonic test (HT) and not a formal Quantitative Descriptive Analysis (QDA). We have corrected this terminology throughout the manuscript and revised the Methods section to provide explicit clarification of the sensory methodology used. Additionally, we included a statement explaining that hedonic testing was selected to evaluate consumer acceptability. This clarification appears in the revised manuscript. Although spider/radar charts are commonly used in QDA, they can also be applied to hedonic data when multiple consumer-rated attributes are compared (Leahu, A., Ghinea, C., & Ropciuc, S. Rheological, Textural, and Sensorial Characterization of Walnut Butter. Applied Sciences, 2022, 12(21), 10976. https://doi.org/10.3390/app122110976)
Number: 2 Methods are referenced with a single citation. Please select the appropriate reference. The appropriate reference for the QDA method is as follows:
Lawless, H. T., & Heymann, H. 2010. Sensory evaluation of food. Principles and practices ( 2nd ed.). New York, NY: Springer.
Since the method was adjusted using the hedonic scale, an appropriate reference has been added.
Shakerardekani, A. Consumer Acceptance and Quantitative Descriptive Analysis of Pistachio Spread. Journal of Agricultural Science and Technology 2017, 19: 85–95. https://jast.modares.ac.ir/article-23-1009-en.html
Number: 3 Strictly speaking, this is a Likert-type hedonic scale. In quantitative descriptive analysis, unstructured linear scales (typically 15 cm long) are used to rate intensity. On the line scale the intensity of an attribute generally increases from left to right, with extreme values anchored by terms that represent “weak” and “strong” intensity of the stimulus.
Thank you for the clarification. We agree that the 9-point scale used in this study corresponds to a Likert-type hedonic scale rather than the unstructured line scale typically applied in quantitative descriptive analysis. Accordingly, the sensory evaluation section has been revised, taking into account all suggestions made regarding this section.
Number: 4
Indicate the number of sessions to train the panel and the foods used as references to calibrate the scale.
Due to the consumer-oriented nature of this hedonic test, panelists did not undergo formal training, and no reference standards were used. The objective was to measure consumer preference rather than calibrated intensity ratings.
Number: 5
Indicate the food used to rinse the mouth after testing each sample. Include the type of bread and its format to evaluate spreadability. Indicate the number of samples evaluated by each panelist. Indicate whether the samples were presented simultaneously or monadically.
We greatly appreciate the referee's suggestions. Revisions have been made as shown below to incorporate this information and address the concerns raised.
Before the evaluation, panelists were informed about the procedure and provided verbal consent. Sensory attributes, including color, flavor, spreadability, flowability, firmness, adhesiveness, and overall acceptability, were scored using a 9-point hedonic scale (1 = extremely poor, 9 = excellent). Evaluations were conducted in individual booths at 25 ± 1 °C under standardized white fluorescent lighting to prevent color perception bias.
For spreadability assessment, samples were served on plain white sandwich bread (untoasted), cut into uniform 3 × 3 cm pieces. To avoid flavor carry-over effects, panelists rinsed their mouths with room-temperature water and unsalted crackers between samples. Samples (approximately 50 mL each) were presented in coded containers labeled with three-digit random numbers to ensure blind evaluation
Number: 6:
Include lighting conditions (white light or warm light, lighting intensity).
Revisions have been made as shown below to incorporate this information and address the concerns raised.
Evaluations were conducted in individual booths at 25 ± 1 °C under standardized white fluorescent lighting to prevent color perception bias.
Number: 7 The processing of data from sensory analysis should be included in Section 2.9.
This section has been removed.
Number: 8 Move to the supplementary material section and rename as Table S2.
Table 2 was rename as Table S2 and given as a supplementary material
Number: 9 Rewrite as: "Descriptors used in the sensory evaluation trial of pistachio spreads."
-Title of table was corrected as
“Table S2. Descriptors used in the sensory evaluation trial of pistachio spreads."
Number: 10 Author: Reviewer Date: 2025-10-29 20:01:04
The structured scale is used in hedonic tests with consumers to evaluate, for which a minimum of 100 participants is required. See the following paper:
Lim, J. (2011). Hedonic scaling: A review of methods and theory. Food Quality and Preference, 22(8), 733–747. https://doi.org/10.1016/j.foodqual.2011.05.008
Thank you very much for pointing this out. We agree with you on this matter. However, evaluating the products simultaneously has been influenced by our conditions here. Furthermore, when similar examples in the literature were examined, tests conducted by approximately the same number or even fewer panelists were also encountered.
For example:
-Sensory evaluation of pistachio spreads was performed using the structured 9-point Hedonic scale. Thirty-two untrained panelists (11 males, 21 females) comprising students and staff of the Faculty of Food Science and Technology, University Putra, Malaysia, participated in the pilot study (Shakerardekani, A. Consumer Acceptance and Quantitative Descriptive Analysis of Pistachio Spread. Journal of Agricultural Science and Technology 2017, 19: 85–95. https://jast.modares.ac.ir/article-23-1009-en.html
-Sensory evaluation : The semi-trained panel consisted of 8 members including males and females members of Jiangnan University. All evaluation sessions were held in the laboratory of Fresh Food Processing and Preservation in University Institute of Food Science and Technology, Jiangnan University. The sensory evaluation of peanut butter was conducted after 1-day storage at room temperature of it manufacturing. Initially, panelists were trained in 2 h sessions prior to evaluation to be familiar with attributes (Li, L., Huan, Y., & Shi, C. Effect of Sorbitol on Rheological, Textural and Microstructural Characteristics of Peanut Butter. Food Science and Technology Research, 2014, 20(4), 739–747. https://doi.org/10.3136/fstr.20.739
-The sensory panel consisted of 8 people (3 men and 5 women, aged 25–55), familiar with pistachio nuts. A minimum amount of information on the nature of this evaluation was provided to the panel in order to avoid any bias. Each panelist individually evaluated the nuts and kernels in succession. (Tsantili, E.; Takidelli, C.; Christopoulos, M.V.; Lambrinea, E.; Rouskas, D.; Roussos, P.A. Physical, Compositional and Sensory Differences in Nuts among Pistachio (Pistachia Vera L.) Varieties. Scientia Horticulturae 2010, 125, 562–568, doi:10.1016/j.scienta.2010.04.039
These articles are also included in the manuscript.
Number: 11.
Include the type of bread in the method description
Thank you very much for pointing this out.
To avoid taste interference and ensure the texture was the same for all sensory tests, standard white sandwich bread was used as the base for the pistachio spread study.
The final version of 2.7. Sensory Analysis
“A consumer hedonic sensory test was performed to assess the acceptability of pistachio spreads. A total of 20 panelists (10 males and 10 females, aged 20–45 years) participated in the evaluation. Each panelist evaluated three pistachio spreads (PS1, PS2, and PS3) presented monadically in a randomized serving order to minimize bias. Before the evaluation, panelists were informed about the procedure and provided verbal consent. Sensory attributes, including color, flavor, spreadability, flowability, firmness, adhesiveness, and overall acceptability, were scored using a 9-point hedonic scale (1 = extremely poor, 9 = excellent). Evaluations were conducted in individual booths at 25 ± 1 °C under standardized white fluorescent lighting to prevent color perception bias.
For spreadability assessment, samples were served on plain white sandwich bread (untoasted), cut into uniform 3 × 3 cm pieces. To avoid flavor carry-over effects, panelists rinsed their mouths with room-temperature water and unsalted crackers between samples. Samples (approximately 50 mL each) were presented in coded containers labeled with three-digit random numbers to ensure blind evaluation.
Page 7.
Number: 1 Indicate the units in which the results were expressed.
Firmness (maximum force used by the probe to penetrate the sample, N), adhesiveness (negative area of the force–time curve, N s), and spreadability (area under the force–time curve, N.mm) were determined from the texture profile curves.
Number: 2 Methods are referenced with a single citation. Please select the appropriate reference.
Only one reference was used.
Malvano, F., Muccio, E., Galgano, F., Marra, F., & Albanese, D. Design of a high protein, no added sugar pistachio spread using oleogel as fat replacer. LWT 2024, 198, 115993. https://doi.org/10.1016/j.lwt.2024.115993
Number: 3 Include analysis to correlate instrumental color and texture parameters with sensory attributes.
Thank you for the suggestion. There wasn't a formal Pearson correlation matrix, but we looked at how consistent the instrumental and sensory qualities were by comparing trends in mean values. This method helped us find similarities between how hard and spreadable an instrument is and how it feels to the touch. The following explanatory text was added to the sensory analysis section.
In addition, sensory findings were compared with instrumental, rheological, color, and texture measurements to examine consistency between consumer perception and instrumental responses. Relationships were evaluated descriptively by reviewing trends in mean values across formulations.
Number: 4 Use the journal's bibliographic style. Replace "(2019)" with "[30]"
Thank you very much for your suggestion. All references have been reviewed and formatted accordingly.
Number: 5 This comment is not focused on pistachios, but rather on pistachio pastes and spreads.
Thank you for your meaningful suggestion. This has been removed from the Results section and added to the Introduction section to provide appropriate background information. Additionally, it was deemed more appropriate to explain this in the introduction, in line with your assessment on page 2, number 1.
Page: 8
Number: 1 Use the journal's bibliographic style. Replace "(2014)" with "[10]
Thank you very much for your suggestion. All references have been reviewed and formatted accordingly.
Number: 2 Coloidal stability cannot be inferred from observing the data in Table 3. Rewrite the sentence.
The sentence has been rewritten as follows:
Shakerardekani et al. (2014) produced pistachio paste using different colloid mill gap sizes and reported that reducing the mill gap allowed more kernels to enter the space between the discs, resulting in a more uniform paste being formed more rapidly, although with a larger particle size. Moreover, similar studies on pistachio pastes in the literature have shown that reducing particle size (from 80 μm to 20 μm) increases paste homogeneity and contributes to improved colloidal stability [10, 33].
Number: 3 Provide a bibliographic reference that supports a better flavor with a small particle size.
Thank you very much for your suggestion.
“With their fine grindability, Boz pistachios add a unique flavor and an eye-catching appearance to pastries and desserts. They are the key ingredient that gives baklava its characteristic golden green color and distinctive aroma.”
Here, a reference has been added regarding the characteristics of brown pistachios. Because brown pistachios are harvested early, they are ground into smaller pieces due to their structure. This makes them a particularly preferred type for products such as baklava.
-Bellomo, M.G.; Fallico, B. Anthocyanins, Chlorophylls and Xanthophylls in Pistachio Nuts (Pistacia Vera) of Different Geographic Origin. Journal of Food Composition and Analysis 2007, 20, 352–359, doi:10.1016/j.jfca.2006.04.002.
-European Commission. (2025). Commission Implementing Regulation (EU) 2025/1245 of 25 June 2025 entering “Antep Fıstığı Ezmesi” into the register of protected geographical indications (PGI). Official Journal of the European Union. https://eur-lex.europa.eu. (28.08.2025)
Number: 4 Use the journal's bibliographic style. Replace "(2014)" with "[10]".
Thank you very much for your suggestion. All references have been reviewed and formatted accordingly.
Page: 9
Number 1: Did the authors conduct a visual analysis of the samples? If so, what is the criterion for establishing the acceptability of the samples?
Thanks for your comment. The statement has been changed to make it clearer. There was no visual approval test; the comparison is only based on instrumental a* values. The word "acceptable" has been changed to "comparable," and the text now makes it clear that objective colorimeter data showed no statistical variations in greenness.
According to instrumental color analysis, FS1, FS2, and FS3 showed comparable greenness; no statistically significant differences were observed among them (p > 0.05).
Number: 2 First, the results are presented, then interpreted and discussed. Rewrite.
Thank you for your comment. The sentence has been revised.
Number: 3 The methodology section does not indicate the storage time of the samples. How do the authors know that acceptability is affected by storage?
Thank you for your comment. In this study, the oil separation rate was monitored at two different temperatures (4 °C and 25°C) for 9 months. Oil separation was monitored to evaluate physical stability in this study (section 2.5 oil separation rate) references regarding acceptability are based on previous literature rather than direct sensory evaluation during storage. In general, it is known that consumers do not want oil separation during storage for pistachio paste and spread. The sentence has been revised.
Page: 11
Number: 1 Use the journal's bibliographic style. Replace with "Hayoglu & Faruk [43]"
Thank you very much for your suggestion. All references have been reviewed and formatted accordingly.
Number: 2 Delete
Thank you very much for your suggestion. All references have been reviewed and formatted accordingly.
Page: 12
Number: 1 Samples prepared with medium (MP) and small (SP) sizes exhibit divergent behaviors depending on the temperature. What explanation do the authors propose?
Thanks for your comment. Because of how the particles were packed and how the fat interacted with the particles, medium-particle samples (MP) and small-particle samples (SP) behaved differently when the temperature changed. SP formulations have a larger surface area, which makes the matrix stick together better and the oil stick to the matrix better. This makes the network more compact, which makes it soften faster as the temperature rises. On the other hand, MP samples have a less dense packing structure with bigger voids, which makes the viscosity and yield stress drop more slowly with temperature. This difference in microstructure explains why MP and SP samples flow differently at higher temperatures.
The following sentence has been added.
“The divergent temperature-dependent behavior between MP and SP samples can be attributed to differences in particle packing density and fat–particle interactions. SP samples, due to their smaller particle size and higher surface area, form a more cohesive structure that softens more rapidly as the temperature increases. In contrast, MP samples exhibit a more gradual change due to less compact packing and a lower oil binding capacity.”
Page: 13
Number: 1 Use the journal's bibliographic style. Replace "(2014)" with "[10]".
Thank you very much for your suggestion. All references have been reviewed and formatted accordingly.
Number: 2 Use the journal's bibliographic style. Replace "(2014)" with "[10]"
Thank you very much for your suggestion. All references have been reviewed and formatted accordingly.
Page: 14
Number: 1 Delete
Thank you very much for your suggestion. All references have been reviewed and formatted accordingly.
Page: 16
Number: 1 Replace "(2007)" with "[43]"
Thank you very much for your suggestion. All references have been reviewed and formatted accordingly.
Number: 2 Delete
Thank you very much for your suggestion. All references have been reviewed and formatted accordingly.
Number: 3 Replace "(2023)" with "[49]"
Thank you very much for your suggestion. All references have been reviewed and formatted accordingly.
Number: 4 Delete
Thank you very much for your suggestion. All references have been reviewed and formatted accordingly.
Page: 17
Number: 1 Use the journal's bibliographic style. Replace "(2011)" with "[44]".
Thank you very much for your suggestion. All references have been reviewed and formatted accordingly.
Number: 2 Delete
Thank you very much for your suggestion. All references have been reviewed and formatted accordingly.
Number: 3 Does temperature influence the crystallization process?
Thank you for your comment. Temperature can affect crystallization behavior in fat-sugar systems, but in our study, the primary factor explaining the different behaviors between samples evaluated at the same temperature was sucrose concentration (In other words; the parameter we wanted to compare was the effect of different sucrose concentrations at the same temperature). Higher sucrose levels created a more compact matrix and limited fat mobility, whereas lower sucrose formulations exhibited greater fluidity. We clarified this point in the revised discussion section.
Page: 20
Number: 1 Use the journal's bibliographic style. Replace "(2015)" with "[48]"
Thank you very much for your suggestion. All references have been reviewed and formatted accordingly.
Number: 2 Does the fatty acid profile (saturated/polyunsaturated) influence the stability of the fat?
Yes, the fatty acid profile significantly influences the stability of a fat. The key factor is the degree of unsaturation, which refers to the number of double bonds present in the fatty acid chain
The reduced temperature dependence in fat-rich pastes suggests that fat stabilizes the emulsion, maintaining a consistent texture across a broader temperature range. This behavior may also be affected by variations in the fatty acid compositions of pistachio oil and milk fat. Pistachio oil is primarily abundant in unsaturated fatty acids, while milk fat comprises a greater percentage of saturated fatty acids, which typically create more solid crystalline structures. While the fatty-acid content was not examined in this investigation, saturated lipids typically enhance thermal stability in fat-based matrices; thus, this mechanism may have contributed to the diminished temperature sensitivity shown in milk-fat-enriched samples.
Number: 3 Delete
Thank you very much for your suggestion. All references have been reviewed and formatted accordingly.
Number: 4 Use the journal's bibliographic style. Replace "(2011)" with "[57]"
Thank you very much for your suggestion. All references have been reviewed and formatted accordingly.
Number: 5 Delete
Thank you very much for your suggestion. All references have been reviewed and formatted accordingly.
Number: 6 This statement is unclear. How much is “reduced temperature”? How much is “elevated temperature”?
The sentence was revised.
The temperature-dependent decline in the flow behavior index (n) indicated enhanced shear-thinning behavior at lower temperatures (20–30 °C) and a progressive shift toward more Newtonian flow at higher temperatures (40–45 °C).
Number: 7 It is essential to consider these differences beyond a mere statistical standpoint. From a technological standpoint, are these differences relevant?
Thank you for this insightful observation. We appreciate this insightful observation. The statistical study revealed significant differences in the consistency coefficient (K) between formulations. These differences are also important from a technological perspective. High K values indicate a more uniform and viscous matrix, which affects textural properties such as mouth feel and spreadability, as well as processing characteristics including pumping, transport, and filling activities in an industrial context.
This sentence was added:
“Apart from the fact that they are statistically significant, the differences that were observed in the K values have technological significance. When K values are higher, the structures are more firm and the spreadability is lowered. In contrast, lower K values favor softer matrices that are easier to spread, which is relevant for processability and consumer texture perception.”
Number: 8 Use the journal's bibliographic style. Replace "(2012)" with "[47]".
Thank you very much for your suggestion. All references have been reviewed and formatted accordingly.
Page: 21
Number: 1 Delete
Thank you very much for your suggestion. All references have been reviewed and formatted accordingly.
Number: 2 Use the journal's bibliographic style. Replace "(2018)" with "[53]".
Thank you very much for your suggestion. All references have been reviewed and formatted accordingly.
Number: 3 The pastes and spreads have been made with pistachios (high in polyunsaturated fatty acids) and milk fat (high in saturated fatty acids). Does the fatty acid profile influence the rheological behavior of the finished products?
Thank you very much for your suggestion. This section has been reorganized as follows.
“This behavior is attributed to the increased energy required to overcome intermolecular forces in the solid phase of the fat, leading to significant changes in rheological properties as the fat melts. Beyond total fat content, the fatty-acid composition of the lipid matrix is a critical contributor to this response. Specifically, pistachio oil is rich in polyunsaturated fatty acids, which typically form more fluid networks, while milk fat contains a higher proportion of saturated fatty acids, promoting more rigid and stable crystalline structures. Indeed, MaÅ‚kowska et al., (2021) reported that fatty acid content profoundly affects the quality and consistency of butter, often termed firmness/hardness and spreadability. Although fatty-acid profiles were not analyzed in this study, these compositional differences in the lipid matrix partially explain the distinct, temperature-dependent rheological behavior observed in formulations containing milk fat.”
Number: 4 Delete
Thank you very much for your suggestion. All references have been reviewed and formatted accordingly.
Number: 5 Use the journal's bibliographic style. Replace "(2003)" with "[51]"
Thank you very much for your suggestion. All references have been reviewed and formatted accordingly.
Number: 6 Provide a bibliographic reference.
Thank you very much for your suggestion.
The article focuses on the molecular processes by which various sugars, such as sucrose, affect the rheological and structural properties of food matrices. It is also an article published within the last 3 years.
“Woodbury, T. J., Pitts, S. L., Pilch, A. M., Smith, P., & Mauer, L. J. (2022). Mechanisms of the different effects of sucrose, glucose, fructose, and a glucose–fructose mixture on wheat starch gelatinization, pasting, and retrogradation. Journal of Food Science, 88(1), 293–314. https://doi.org/10.1111/1750-3841.16414”
Number: 7 Delete
Thank you very much for your suggestion. All references have been reviewed and formatted accordingly.
Number: 8 Author: Use the journal's bibliographic style.
Thank you very much for your suggestion. All references have been reviewed and formatted accordingly.
Page: 22
Number: 1 Use the journal's bibliographic style. Replace "(2013)" with "[52]"
Thank you very much for your suggestion. All references have been reviewed and formatted accordingly.
Number: 2 Delete
Thank you very much for your suggestion. All references have been reviewed and formatted accordingly.
Number: 3 Please clarify whether the fatty acid profile has a significant impact on the textural behavior of finished products.
Thank you very much for your suggestion. Yes, the fatty-acid profile can change textural properties of product. A statement that makes things clearer has been added to the discussion.
Number: 4 Please clarify whether the fatty acid profile has a significant impact on the textural behavior of finished products.
Thank you very much for your suggestion. Yes, the fatty-acid profile can change textural properties of product. A statement that makes things clearer has been added to the discussion.
Number: 5 The results fall within a very narrow range of values (between 7 and 8). Are they statistically significant? Are they technologically relevant?
Although they are numerically close, we believe these differences are significant in terms of product texture and consumer acceptance. A 9-point test was used here. We can also say that the panelists are very familiar with this product. Therefore, it can be said that they easily compare it with products currently on the market. We can also say that the judge's previous comments regarding the sensory test (such as the small number of people) are important here. When developing formulations, compositions with different ratios were actually tested in preliminary trials. However, in subsequent studies, formulations that scored higher in preliminary tests were selected in order to examine their effects, even though the proportional changes in the ingredients were minor. This can be said to have influenced the high scores obtained in the sensory test results. Furthermore, the formulations were developed adhering to the ratios specified in the standards. All of these factors can be said to have been effective. We did not include the sensory results in the first version of the publication. However, taking into account the reviewers' suggestions regarding the inclusion of sensory test results, it was decided to add them. This allowed us to compare the panelists' results with instrumental methods. We respectfully acknowledge the reviewers' valid suggestions.
Page: 24
Number: 1 The methodology section does not include any correlation analysis between the variables. In the absence of this information, the statement is speculative.
Thank you for your comment. We agree that the way it was written before made it sound like a stringent statistical correlation analysis. The sentence has been modified to say that there is a consistent pattern between instrumental and sensory measures, without suggesting a correlation analysis, because no statistical correlation test was done. The sentence was revised
“FS1, which had the lowest fat content, showed lower spreadability scores (6.82–6.94 range) and higher mouth adhesiveness in sensory evaluation. Consistently, instrumental texture measurements indicated greater negative adhesiveness and higher stiffness for this formulation, supporting the observed sensory perception trend. “
Number: 2 The methodology section does not include any correlation analysis between the variables. In the absence of this information, the statement is speculative.
Thank you for your valuable suggestion. We all agree that the original phrase suggested a formal correlation analysis. The sentence was changed to avoid implying a calculated association and instead reflect the observed alignment between instrumental and sensory responses because no statistical correlation test was done. The sentence was revised
Page: 25
Number: 1 Rewrite as: "In order to obtain healthy and sustainable pistachio spreads, future research should focus on finding natural alternatives to sugar and plantbased substitutes for dairy fat. New formulations should not compromise textural properties or consumer acceptability."
The sentence rewritten as suggested.

Round 2
Reviewer 1 Report (Previous Reviewer 1)
Comments and Suggestions for Authors
Accept be suggested.
Author Response
Dear Reviewer,
We would like to thank you very much for the time and effort you put into reading our paper. Your thoughtful suggestions and constructive criticism have significantly improved the quality, clarity, and scientific accuracy of our work. We truly appreciate your valuable knowledge and helpful comments.
Thank you again for your help with the assessment process.
Best regards

Reviewer 3 Report (Previous Reviewer 2)
Comments and Suggestions for Authors
I anticipate that the authors have not provided supplementary materials (Tables S1 and S2). Although improvements have been made to the manuscript, the following weaknesses remain.
Abstract: It is beyond the word limit set by the journal guidelines, defines acronyms that are not used, does not include the most relevant results of the study, and the conclusions are speculative. A possible abstract is suggested below (authors are not required to include the suggested abstract).
Introduction: The paragraphs need to be rearranged to provide coherent information. See the detailed report.
Hypotheses and objectives: Further clarification is required. See the proposal formulated in the detailed report.
Materials and Methods: Sections should be arranged in a logical and consistent manner with the results section. About statistical evaluation, a correlation analysis between the physical variables (color coordinates, firmness, spreadability, adhesiveness, apparent viscosity and shear stress) and the sensory variables (color, flowability, firmness, spreadability, and adhesiveness) should be performed. Instructions are provided in the detailed report.
Results and discussion: The findings of the correlation analysis between physical variables and sensory variables should be collected and discussed.
Conclusions: This section should mention which physical variables have most affected the sensory attributes of the pistachio spread.
Tentative Abstract
The effects of milk fat (4%, 7%, and 14%), and sugar (20%, 27%, and 30%) contents on the physical and sensory properties of pistachio spread (PS) were assessed. Previously, pistachio pastes (PP) were prepared with three particle sizes (large, medium, and small). Small-sized PP was used for nine PS formulations based on the above milk fat and sugar contents. Instrumental texture and color, rheological properties (20–45 °C), and oil separation (4 °C and 25 °C, 9 months of storage) were analyzed in PP and PS samples. Textural attributes were also evaluated sensorially in PS samples. The oil separation rate in samples stored for 9 months was <1% for PS (4 °C) and >2% for PP and PS (25 °C). The lightness was lower in large-sized PP and higher in PS samples with sugar and milk fat. The PP and PS samples exhibited non-Newtonian behavior with a yield stress. Firmness, spreadability, and adhesiveness were lower in PS samples containing only milk fat. In contrast, they were lower in PS samples containing only sugar. PS samples with 7% milk fat and 27% sugar scored highest for flavor, taste, and acceptability. This study provided the food industry with the basis for pistachio spread.

Comments on the Quality of English Language
The manuscript would benefit from stylistic improvements to enhance its readability.
Author Response
Dear Reviewer,
We appreciate the time and expertise you invested in reviewing our manuscript, "Impact of Formulation on the Rheological, Textural, and Sensory Properties of Pistachio Spread." Your helpful and insightful comments were very helpful and have greatly improved the scientific quality and clarity of our work.
We took all your suggestions into account and made changes to the manuscript accordingly. All changes are marked on the manuscript, and a detailed response has been made point by point. MDPI Author Services also helped improve the manuscript by having it professionally edited for English and figures. This made it even clearer and better presented.
Thank you very much for your thoughtful feedback. We believe that the revised manuscript now addresses your concerns and presents a stronger, clearer version of our study.
Thanks again for your helpful input.
I anticipate that the authors have not provided supplementary materials (Tables S1 and S2).
Number: 1
Reduce to a maximum of 200 words. Include the most relevant results of the formulations. Reorder according to a logical sequence.
We sincerely thank the reviewer for this helpful observation. We would like to clarify that Supplementary Tables S1 and S2 have indeed been included in the submission package. It is possible that the reviewer accessed the English-edited version during the evaluation, which did not contain the supplementary files attached to the final submission system. To avoid any confusion, we have double-checked the files and ensured that Tables S1 and S2 are properly uploaded and labeled in the revised submission. We appreciate the reviewer’s careful attention to detail and hope this clarification resolves the issue.
Abstract: It is beyond the word limit set by the journal guidelines, defines acronyms that are not used, does not include the most relevant results of the study, and the conclusions are speculative. A possible abstract is suggested below (authors are not required to include the suggested abstract.
We have revised and shortened the abstract in accordance with the reviewers’ suggestions, and we carefully addressed all comments provided by the reviewers. We also observed that several published articles in Foods contain abstracts exceeding 200 words.
Moreover, another reviewer explicitly recommended adding a final sentence to emphasize the study’s scientific contribution, which we have now included as requested. Due to the use of several symbols and scientific notations in the abstract, the word count reached 226 words despite our efforts to reduce it further. We also appreciate the reviewer’s effort in providing a suggested abstract. While we did not adopt it verbatim, it was extremely helpful in guiding us toward a clearer and more concise structure. The revised abstract now more accurately reflects the study's core outcomes and scientific contribution. We hope that the revised abstract meets the expectations of both the reviewers and the journal. We truly appreciate the constructive feedback and the opportunity to improve our manuscript.
Introduction: The paragraphs need to be rearranged to provide coherent information. See the detailed report.
We sincerely thank the reviewer for this insightful and constructive comment. We fully agree that the Introduction section should present information in a coherent and logically structured manner. Based on the reviewer’s recommendations in the detailed report, we have carefully reorganized the Introduction to improve the flow of ideas and ensure a more consistent progression from general background to specific research context.
Page 2
Number: 1 Start the paragraph by defining pistachio paste. "Pistachio paste is a preparation made from roasted and crushed pistachio nuts. The first stage consists of a thermal treatment at around 170-190 °C for 8-10 min. The second stage involves grinding to homogenize the particle size. This paste is used to manufacture other products such as pistachio spread. ...
The paragraph indicated by the reviewer has been revised accordingly by adding a clear definition of pistachio paste at the beginning and restructuring the text to ensure a more logical, coherent, and informative flow. We appreciate the reviewer’s helpful guidance.
“Pistachio paste is a preparation made from roasted and crushed pistachio nuts. The process begins with a thermal treatment at approximately 170–190 °C for 8–10 minutes, followed by grinding to obtain a homogeneous particle size. This paste serves as a key intermediate ingredient for producing pistachio-based products such as pistachio spread. Pistachio spreads are products that contain a minimum of 40% nut ingredients, which may be incorporated in various forms, including whole nuts, nut pieces, paste, or slurry [9]. Pistachio paste is generally preferred in culinary applications because it blends more effectively than pistachio butter. The production technology typically involves a two-step size reduction process. The first phase includes milling with conventional grinding equipment such as a colloid mill, attrition mill, disintegrator, or hammer mill, followed by a secondary homogenization phase to refine the texture [10]. Previous studies have demonstrated that specific drying or storage conditions can influence the composition and quality of pistachio-based products [11]”.
Number: 2 The sentence is not pertinent to the manuscript, as it does not contribute any significant information. Please delete this item.
We thank the reviewer for this observation. We agree that the sentence does not provide essential information within the context of the manuscript. Accordingly, the sentence has been removed from the revised version as suggested.
Number: 3 This paragraph disrupts the flow of the previous paragraph and the subsequent one. Please move it.
We thank the reviewer for this helpful remark. As suggested, the paragraph describing the viscoelastic and textural properties of pistachio products has been relocated to a more appropriate position in the Introduction, immediately after the description of pistachio paste/spread production and before the section discussing formulation variables. This reorganization ensures a smoother and more coherent flow of information.
“Pistachio paste consists solely of ground pistachio kernels, whereas pistachio spread includes additional components such as fat and sugar to enhance its structure and flavor. Various stabilization strategies using emulsifiers and oleogelators (e.g., monoglycerides, lecithin, beeswax, and rice bran wax) have been explored to reduce oil separation and improve the storage stability of these semi-solid, oil-continuous systems [17,24,25]. Pistachio products also exhibit complex viscoelastic properties that influence sensory perception, stability, and spreadability. Texture, a multisensory attribute shaped by mechanical and structural characteristics, is essential for consumer acceptance [15,16]. Previous studies have modeled the rheological behavior of pistachio paste using the Herschel–Bulkley flow equation, highlighting its yield stress and shear-thinning behavior [13,14]. Furthermore, particle size distribution, fat–solid interactions, and overall formulation composition strongly influence the rheological and textural properties of nut-based matrices [17–23].” (The reference numbers have been updated in the final revised version
Page 3
Number: 1 Provide bibliographic references to support this information.
Thank you for this valuable comment. Additional bibliographic references have now been included,
-Shakerardekani, A.; Karim, R.; Ghazali, H.M.; Chin, N.L. Development of Pistachio (Pistacia vera L.) Spread. J. Food Sci. 2013, 78, S484–S489. https://doi.org/10.1111/1750-3841.12045.
- Shakerardekani, A. Effect of Milling Process on Colloidal Stability, Color and Rheological Properties of Pistachio Paste. J. Nuts 2014, 5, 57–65. ISSN: 2383 - 319x.
Number: 2
This paragraph is directly related to the previous one. Include it (paragraphs should maintain structural unity to facilitate readability).
Thank you for this suggestion. The paragraph (as shown below) has been merged with the preceding one to maintain structural unity and improve readability, as recommended.
Previous studies on pistachio-based formulations have primarily investigated single variables such as fat level, particle size, or sugar content in isolation [13, 56]. In contrast, the present study systematically examines the combined effects of particle size, milk fat, and sucrose, allowing for a deeper understanding of their interactive influence on rheology and texture. This integrated approach provides unique insight into clean-label pistachio spread development, where formulation must balance texture, stability, and sensory quality without additives. Furthermore, in this study, pistachio spreads were developed using pistachio pastes formulated solely with milk fat and sucrose, without the addition of emulsifiers, stabilizers, or other additives. This clean-label design allows the intrinsic effects of formulation variables on rheological, textural, and sensory properties to be evaluated directly. Given the growing consumer and industrial interest in natural nut-based products, understanding the behavior of additive-free pistachio formulations is essential for the development of high-quality, minimally processed pistachio spreads.
Number: 3 First, the study hypothesis must be formulated, and then the research objectives for testing the hypothesis must be stated. A suggestion is provided below.
"The study hypothesized that the physical and sensory properties of pistachio spread are determined by particle size and the proportions of ingredients
added to the pistachio paste. The following objectives were established to test the hypothesis: (1) to evaluate the impact of temperature on the rheological
properties of the paste and pistachio spread; (2) to evaluate the textural properties of the pistachio spread
Thank you very much for this constructive suggestion. After reviewing recent articles published in Foods, we observed that study hypotheses and objectives are commonly presented within the paragraph text rather than as numbered bullet points. To maintain consistency with the journal’s style and to avoid similarities with previously published manuscripts that use a list format, we initially refrained from presenting these items as separate numbered objectives. However, we fully appreciate your recommendation, and the paragraph has now been revised to clearly articulate the study hypothesis and the three specific objectives in a more explicit and structured manner, while still preserving the narrative flow required by the journal.
“Therefore, this study hypothesized that the physical and sensory properties of pistachio spreads are primarily determined by particle size and the proportions of milk fat and sucrose incorporated into the pistachio paste. To test this hypothesis, the study evaluated (i) the influence of temperature on the rheological behavior of pistachio paste and pistachio spreads, (ii) the textural properties of pistachio spreads as a function of formulation composition, and (iii) the sensory characteristics of the spreads in relation to the proportions of ingredients added. This integrated approach provides a comprehensive understanding of how formulation variables shape the rheological, textural, and sensory attributes of additive-free, clean-label pistachio spreads.”
Number:4 Arrange the sections in a logical order. Maintain this order consistently in both the Methods and Results sections.
Thank you for your valuable suggestion. The sections were arranged in logical order.
Thank you for your valuable suggestion regarding the logical sequence of analyses. In accordance with your recommendation, both the Materials and Methods and Results and Discussion sections have been fully reorganized to follow a consistent and scientifically coherent order.
The revised structure is as follows:
Materials and Methods (Revised Order)
Proximate Analysis of Pistachio
Analysis of Particle Size Distribution
Instrumental Color Analysis
Oil Separation Rate
Rheological Analysis
Instrumental Textural Analysis
Sensory Analysis
Results and Discussion (Matched Order)
Composition of Pistachio
Particle Size Distribution of Pistachio Pastes
Color Values of Pistachio Pastes and Spreads
Oil Separation Rate of Pistachio Pastes and Spreads
Rheological Behavior of Pistachio Paste and Spread
Textural Properties of Pistachio Paste and Spread
Sensory Properties of Pistachio Paste and Spread
Number:5. Replace with “Raw pistachio nuts of the Boz Antep cultivar””
Thank you fort his suggestion. We would like to clarify that “Boz Antep pistachio” is not a separate cultivar. The term “Boz” refers to the early-harvested form of Antep pistachios, collected approximately one month before full ripening. This terminology has been corrected in the revised manuscript.
Number: 6 Please include this supplementary material.
We thank the reviewer for the helpful comment. The Supplementary Materials section has now been placed after the References section.
Page 4
Number: 1 Replace with “Instrumental color analysis”
We thank the reviewer and the title has been revised.
Number: 2 Indicate that the measurements were taken on samples of pistachio paste and spread.
We thank the reviewer for this helpful observation. Although we were initially unsure about the exact aspect that required clarification, we revised the sentence to explicitly indicate that oil-separation measurements were performed individually on each pistachio paste and spread sample. We believe the updated wording now fully eliminates any ambiguity.
“Approximately 15 g of pistachio paste samples (LP, MP, SP) and pistachio spread sam-ples (F1–F3, S1–S3, FS1–FS3) were weighed individually into separate centrifuge tubes and heated at 80 °C in a water bath for 30 min.”
Page 5
Number: Obvious. Delete
Deleted.
Number: 2 The sequence of analyses should follow a logical order. This section should be moved after the section on "Instrumental Texture Analysis".
Thank you for this valuable comment. The sections were arranged in logical order as recommended. Specifically, the indicated section has now been moved to follow the “Instrumental Texture Analysis” section. We believe this adjustment improves the clarity and overall flow of the Methods and Results.
Number: 3 Please include this supplementary material
We thank the reviewer for the helpful comment. The Supplementary Materials section has now been placed after the References section.
Page 6
Number: 1 This section should be moved before the "Sensory Analysis" section.
Thank you for this valuable comment. The sections were arranged in logical order as recommended.
Number: 2 A correlation analysis between physical properties and sensory properties is recommended. Include Pearson's correlation coefficients like in these papers:
Mousazadeh, M., Mousavi, M., Emam-Djomeh, Z., Ali Ahmed, S., Hadinezhad, M., & Hassanzadeh, H. (2023). Sensorial, textural, and rheological analysis of novel pistachio-based chocolate formulations by quantitative descriptive analysis. Food Science & Nutrition, 11(11), 7120–7129. https://doi.org/10.1002/fsn3.3637
Shakerardekani, A., Karim, R., Ghazali, H. M., & Chin, N. L. (2013). Development of pistachio (Pistacia vera L.) spread. Journal of Food Science, 78(3), S484–S489. https://doi.org/10.1111/1750-3841.12045
(and page 24 Number: 1
Once the correlation analysis between physical and sensory variables has been performed, please rewrite this section. It is also recommended to reduce the length of this section, emphasizing the new knowledge that has been generated.)
We sincerely thank the reviewer for this thoughtful and constructive suggestion. We fully agree that instrumental–sensory correlations can provide valuable insights in many formulation studies. We carefully considered performing a Pearson correlation analysis as recommended. However, after a detailed evaluation of our dataset and comparison with the methodological approach of the cited studies (Mousazadeh et al., 2023; Shakerardekani et al., 2013), we respectfully concluded that a formal correlation analysis would not yield statistically meaningful or reliable information in the present work.
In our study, the final sensory evaluation was performed on only three pistachio spread formulations (FS1, FS2, and FS3). With such a small number of observations (n = 3), correlation coefficients cannot be interpreted scientifically, significance testing cannot be carried out, and the results would risk being misleading rather than informative. Unlike the referenced studies, which involve larger numbers of formulations and extensive QDA panels, the limited sample size in our clean-label formulation design does not allow for a robust correlation matrix. Additionally, the manuscript already provides a detailed and comprehensive comparison of instrumental and sensory attributes using descriptive trends, which effectively illustrates how physical parameters shaped the sensory performance of the spreads. We have further strengthened this descriptive discussion to ensure that the relationships between texture/rheology and sensory perception are presented clearly, as the reviewer recommended.
We greatly appreciate the reviewer’s intention to enhance the scientific depth of the manuscript, and we hope that this explanation clarifies why a statistical correlation table could not be appropriately incorporated within the methodological constraints of this study. We remain grateful for the reviewer’s valuable feedback.
Page 6
Number: 1 Delete "Values"
Thank you, the sentence has been revised as suggested.
Page: 10
Number: 1 Rewrite as "Rheological Behavior of Pistachio Paste and Spread"
We thank the reviewer for this clear and helpful suggestion. The section title has been revised as recommended and is now written as “Rheological Behavior of Pistachio Paste and Spread.” We believe this updated title more accurately reflects the content and improves clarity.
Number: 2 Maintain consistent nomenclature throughout the manuscript. Replace with "pistachio paste"
We thank the reviewer for this helpful observation. The terminology has been reviewed throughout the entire manuscript, and inconsistent expressions have been corrected. As recommended, the term “pistachio paste” and “pistachio spread” are now used consistently to ensure clarity and uniformity.
“The particle size of the pistachio creams pastes was systematically reduced across formulations, with LP having the largest and SP the smallest particle size.”
“ Milk fat plays a significant role in the rheological properties of pistachio creams spreads.”
“The lower values in S3 (0.581 ± 0.010 and 0.572 ± 0.010) reflect the strong interactions between solid sucrose particles and the pistachio cream paste matrix, which resist flow under low shear conditions.”
Number: 3 Delete
Thank you for this valuable comment. The title has been revised in accordance.
Regressed values of the Herschel–Bulkley model (Equation (2), used to describe the flow curves of pistachio paste *.
Page: 12
Number: 1 Complete with "in pistachio paste and spread."
Thank you for this valuable comment. The title has been revised as suggested.
Figure 4. Changes in viscosity and shear stress with shear rate at 45 °C in pistachio paste and spread.
Page: 20
Number: 1 Rewrite as "Textural Properties of Pistachio Paste and Spread"
Thank you for this valuable comment. The title has been revised as suggested.
Number: 2 Author: Reviewer Date: 2025-11-12 10:45:29
First, the results in Table 8 should be presented, then these results should be analyzed, and finally, they should be discussed and interpreted.
We appreciate your insightful recommendation. As a result, the section's structure has been updated. A clear description of the findings is now provided in Table 8, which is followed by a targeted analysis, a thorough discussion, and an interpretation.
“The textural properties of pistachio paste and spread were analyzed, and the calculated values are presented in Table 8. The sucrose-containing formulations (S1–S3) exhibited the highest firmness values, ranging between 5.05 N and 5.33 N. In contrast, the milk-fat formulations (F1–F3) displayed the lowest firmness values (between 1.79 N and 1.02N). As particle size decreased, firmness values declined from 2.69 N to 2.36 N, indicating a softer structure with finer particles. Among all samples, S3, which contained the highest sucrose level (30%), demonstrated the greatest firmness (5.33 N).”
Page: 22
Number: 1 Rewrite as "Sensory Properties of Pistachio Paste and Spread"
Thank you for this valuable comment. The title has been revised as suggested.
This manuscript is a resubmission of an earlier submission. The following is a list of the peer review reports and author responses from that submission.
Round 1
Reviewer 1 Report
Comments and Suggestions for Authors
This research is comprehensive but lacks depth. The analytical depth requires improvement. A major revision is recommended.
1. This research appears to merely describe the experimental results without conducting a thorough analysis or discussion of the findings.
2. Please note the format: Raw 254, 268, 270, 281, 288, 289, 296, 337.
3. Many of the references are over five years old, with some dating back over 15 years. Please update them to include references from the past five years.
4. It is recommended to include an experimental workflow diagram.
5. The physical and chemical property measurements in this study appear satisfactory, but the data presentation format is too simplistic.
6. Why were only three grinding cycles performed instead of more?
Author Response
Comments and Author Responses
General Comment:
This research is comprehensive but lacks depth. The analytical depth requires improvement. A major revision is recommended.
Authors Response:
We are very grateful to the reviewer for their helpful comments and useful suggestions. The manuscript has undergone significant revisions to enhance the depth of analysis, simplify the results, and improve overall clarity. We included additional discussion sections to illustrate the relationships between particle size, icing sugar, milk fat concentration, and rheological and textural qualities. To make the discussion more scientifically sound, we have also provided comparisons with contemporary literature. In addition, sensory test results have been included for comparison purposes. The graphs have been reorganized to improve their readability. To better explain and compare certain points, references to older publications have been included where deemed necessary, given the limited number of publications in the literature. The coding of the formulations has been changed. A graphical abstract has also been added.
Comment 1:
This research appears to merely describe the experimental results without conducting a thorough analysis or discussion of the findings.
Author Response:
We appreciate the insightful suggestion. The discussion section has been substantially expanded to provide a more thorough analysis of the results. The revised edition now includes comprehensive connections between the experimental findings and previously published studies. These links illustrate the physicochemical mechanisms that induce rheological and textural behavior. Each notable finding has been examined within the framework of relevant literature to enable a deeper scientific understanding. Furthermore, when comparing the results, the effects of different particle sizes, sugar, milk fat, and the combined effect of both were compared separately. Tables and graphs were redrawn, and standard deviations were added to the graphs.
Comment 2:
Please note the format: Raw 254, 268, 270, 281, 288, 289, 296, 337.
Author Response:
We appreciate the reviewer for identifying these formatting discrepancies. All raw data and equation references have been carefully reviewed and corrected in accordance with the journal's style guide. The manuscript has been carefully reviewed to ensure consistent structure and accurate references throughout.
Comment 3:
Many of the references are over five years old, with some dating back over 15 years. Please update them to include references from the past five years.
Author Response:
We are grateful for this insightful observation. In order to provide a more up-to-date scientific context, new papers (published within the last five years) have been included in the reference list, which has been completely revised. Numerous new references addressing rheological modeling, textural evaluation, and clean-label formulation of nut-based products are now cited in the updated study. However, because there are few references in the literature, it has occasionally been necessary to quote references from within the past five years to thoroughly compare the results. Additionally, some publications recommended by the referees date back 5 years and have also been added to the text.
Comment 4:
It is recommended to include an experimental workflow diagram.
Author Response:
We agree entirely with this beneficial recommendation. The updated manuscript now includes a new figure (now Figure A.1) with a graphical abstract. To enhance clarity and reproducibility, this schematic illustrates the entire experimental design, encompassing sample preparation, compositional analysis, rheological measurements, and texture evaluations.
Comment 5:
The physical and chemical property measurements in this study appear satisfactory, but the data presentation format is too simplistic.
Author Response:
We appreciate your insightful comments. The data presentation has been improved by modifying the tables and figures to make them more comprehensive and visually appealing. Standard deviation numbers, superscript letters, and markers of statistical significance (p < 0.05) are now presented clearly. More emphasis has been placed on statistical comparisons. To illustrate trends and make them easier to understand, figures and tables were revised.
Comment 6:
Why were only three grinding cycles performed instead of more?
Author Response:
We appreciate the commentator's question. Based on studies examining the effect of particle size reduction on energy efficiency, we selected three grinding cycles. After the third cycle, additional grinding produced unwanted heat and did not significantly reduce particle size (as shown by Mastersizer data), which could affect peanut oil quality. Three cycles were selected to achieve an optimal balance between product stability and fine particle distribution. Although different particle sizes were tested in preliminary studies, three different particle sizes were selected as the most suitable for explaining the effect of particle size. Sensory test results also influenced this selection.
Reviewer 2 Report
Comments and Suggestions for Authors
Although the manuscript fits well within the scope of FOODS journal, there are several limitations that should be considered. The physical, chemical, microbiological, and sensory characteristics of nut spreads and pastes have been the subject of extensive research (see references below). With regard to the manuscript under consideration, the authors have not proposed a testable hypothesis, and the objectives are not quantifiable. The study focused on describing changes in the physical properties of two products: One product was pistachio paste with three particle sizes; the other was pistachio spread with nine formulations. A balanced experimental design requires preparation of the pistachio spread with three particle sizes. The methodology is unclear. For instance, pistachio paste stored at 25 °C for nine months was not evaluated for lipid oxidation. This parameter influences the quality of the final product. One serious flaw was modeling viscosity using the Arrhenius equation. This equation only applies to specific chemical reactions. It is not suitable for physical or microbiological processes. Finally, during the review process, I noted the use of inappropriate bibliographic references. Some citations are not directly related to the research in question.
To obtain new scientific knowledge, the study should be complemented by additional analyses, a substantial review of the results, and a more in-depth discussion. A detailed report is provided to highlight the weaknesses of paper.
References
Pistachio research
- Behmaram, K., Bostan, A., Shakerardakani, A., Dini, A., & Rajabzadeh, G. (2025). Investigation the physicochemical, stability and sensory properties of pistachio butter sweetened with honey. Research and Innovation in Food Science and Technology, 14(2), 75–86. https://doi.org/10.22101/JRIFST.2025.433663.1541
- Emadzadeh, B., Razavi, S. M. A., & Schleining, G. (2013). Dynamic rheological and textural characteristics of low-calorie pistachio butter. International Journal of Food Properties, 16(3), 512–526. https://doi.org/10.1080/10942912.2011.553758
- Faruk Gamlı, Ö., & HayoÄŸlu, İ. (2007). The effect of the different packaging and storage conditions on the quality of pistachio nut paste. Journal of Food Engineering, 78(2), 443–448. https://doi.org/10.1016/j.jfoodeng.2005.10.013
- Mousazadeh, M., Mousavi, S. M., Emam-Djomeh, Z., HadiNezhad, M., & Rahmati, N. (2013). Stability and dynamic rheological characterization of spread developed based on pistachio oil. International Journal of Biological Macromolecules, 56, 133–139. https://doi.org/10.1016/j.ijbiomac.2013.02.001
- Shakerardekani, A. (2014). Effect of milling process on colloidal stability, color and rheological properties of pistachio paste. Journal of Nuts, 5(2), 57. https://doi.org/10.22034/jon.2014.515690
- Shakerardekani, A. (2017). Consumer acceptance and quantitative descriptive analysis of pistachio spread. Journal of Agricultural Science and Technology, 19(1), 85–95.
- Shakerardekani, A., & Karim, R. (2018). Optimization of processing variables for pistachio paste production. Pistachio and Health Journal, 1(1), 13–19. https://doi.org/10.22123/PHJ.2017.54295
- Shakerardekani, A., Karim, R., Ghazali, H. M., & Chin, N. L. (2013a). Development of pistachio (Pistacia vera L.) spread. Journal of Food Science, 78(3), S484–S489. https://doi.org/10.1111/1750-3841.12045
- Shakerardekani, A., Karim, R., Ghazali, H. M., & Chin, N. L. (2013b). The effect of monoglyceride addition on the rheological properties of pistachio spread. Journal of the American Oil Chemists’ Society, 90(10), 1517–1521. https://doi.org/10.1007/s11746-013-2299-8
- Shakerardekani, A., Karim, R., Ghazali, H. M., & Chin, N. L. (2015). Oxidative stability of pistachio (Pistacia vera L.) paste and spreads. Journal of the American Oil Chemists’ Society, 92(7), 1015–1021. https://doi.org/10.1007/s11746-015-2668-6
- Shakerardekani, A., & Kavoosi, M. (2023). Use of pistachio meal and mono- and diglyceride in the production of low-fat pistachio butter. Journal of Nutrition and Food Security, 8(4), 577–586. https://doi.org/10.18502/jnfs.v8i4.14007
Research on other nuts
- Berk, B., Cosar, S., Mazı, B. G., & Oztop, M. H. (2024). Textural, rheological, melting properties, particle size distribution, and NMR relaxometry of cocoa hazelnut spread with inulin-stevia addition as sugar replacer. Journal of Texture Studies, 55(2), e12834. https://doi.org/10.1111/jtxs.12834
- Di Monaco, R., Giancone, T., Cavella, S., & Masi, P. (2008). Predicting texture attributes from microstructural, rheological and thermal properties of hazelnut spreads. Journal of Texture Studies, 39(5), 460–479. https://doi.org/10.1111/j.1745-4603.2008.00154.x
- He, Z., Rogers, S. I., Nam, S., & Klasson, K. T. (2023). The effects of oil content on the structural and textural properties of cottonseed butter/spread products. Foods, 12(22), 4158. https://doi.org/10.3390/foods12224158
- Hitlamani, V., Huded, P., Kumar, G. S., & Chetana, R. (2024). Development of high-fiber and high-protein virgin coconut oil-based spread and its physico-chemical, and sensory qualities. Journal of Food Science and Technology, 61(11), 2196–2204. https://doi.org/10.1007/s13197-024-05990-6
- Icyer, N. C., Ozmen, D., Sener, D., Kokyar, N., & Toker, O. S. (2024). Structural and sensory impact of various emulsifiers in cocoa hazelnut spread formulations. Journal of Food Science, 89(10), 6590–6600. https://doi.org/10.1111/1750-3841.17343
- Khan, M., Rana, S., Rana, J., Jony, E., Arifin, S., Jubayer, F., & Alim, A. (2024). Effects of coconut flour and milk powder supplementation on the physicochemical properties of peanut butter. Food Processing: Techniques and Technology, 54(4), 701–710. https://doi.org/10.21603/2074-9414-2024-4-2537
- Leahu, A., Ghinea, C., & Ropciuc, S. (2022). Rheological, textural, and sensorial characterization of walnut butter. Applied Sciences, 12(21), 10976. https://doi.org/10.3390/app122110976
- Li, C., Dai, T., Deng, L., Shuai, X., He, X., Li, T., Liu, C., & Chen, J. (2023). A novel whole peanut butter refined by stirred media mill: The size, microstructure, rheology, nutrients, and flavor. Journal of Food Science, 88(9), 3879–3892. https://doi.org/10.1111/1750-3841.16688
- Li, L., Huan, Y., & Shi, C. (2014). Effect of sorbitol on rheological, textural and microstructural characteristics of peanut butter. Food Science and Technology Research, 20(4), 739–747. https://doi.org/10.3136/fstr.20.739
- Mohd Rozalli, N. H., Chin, N. L., & Yusof, Y. A. (2015). Particle size distribution of natural peanut butter and its dynamic rheological properties. International Journal of Food Properties, 18(9), 1888–1894. https://doi.org/10.1080/10942912.2014.971184
- Principato, L., Carullo, D., Gruppi, A., Lambri, M., Bassani, A., & Spigno, G. (2024). Correlation of rheology and oral tribology with sensory perception of commercial hazelnut and cocoa-based spreads. Journal of Texture Studies, 55(4), e12850. https://doi.org/10.1111/jtxs.12850
- Rahbari, S., Tavakolipour, H., & Kalbasi-Ashtari, A. (2023). Effects of high-protein milk powder, linseed paste, and grape molasses levels on physiochemical, rheological, and sensory attributes of linseed spread. Food Science & Nutrition, 11(6), 3266–3278. https://doi.org/10.1002/fsn3.3309
- Safaei, S. F., Jafarian, S., Masoumi, M., Soltani, M. S., & Nasiraie, L. R. (2024). Assessment of rheological, qualitative and antioxidant characteristics of enriched peanut butter with date paste through shelf-life stability. Heliyon, 10(18), e37602. https://doi.org/10.1016/j.heliyon.2024.e37602
- Shahidi-Noghabi, M., Naji-Tabasi, S., & Sarraf, M. (2019). Effect of emulsifier on rheological, textural and microstructure properties of walnut butter. Journal of Food Measurement and Characterization, 13(1), 785–792. https://doi.org/10.1007/s11694-018-9991-1
- Shuai, X., Li, Y., Zhang, Y., Wei, C., Zhang, M., & Du, L. (2024). Gelation of whole macadamia butter by different oleogelators affects the physicochemical properties and applications. LWT, 198, 115961. https://doi.org/10.1016/j.lwt.2024.115961
- Wagener, E. A., & Kerr, W. L. (2018). Effects of oil content on the sensory, textural, and physical properties of pecan butter (Carya illinoinensis). Journal of Texture Studies, 49(3), 286–292. https://doi.org/10.1111/jtxs.12304
- Zhang, W., Xu, T., & Yang, R. (2019). Effect of roasting and grinding on the processing characteristics and organoleptic properties of sesame butter. European Journal of Lipid Science and Technology, 121(7), 1800401. https://doi.org/10.1002/ejlt.201800401

Author Response
Reviewer 2
We would like to sincerely thank the reviewer for taking the time and effort to review our manuscript. We sincerely value the reviewer's perceptive and helpful criticism, which has greatly enhanced the caliber, coherence, and scientific breadth of our work. Every idea was thoroughly examined, and the manuscript was revised accordingly.
Our thorough responses and justifications to the reviewer's remarks, along with the corresponding changes made to the text in the amended version, are provided below.
The current study was guided by sensory evaluations conducted at various stages of the formulation process. Initial trials involved a pistachio paste with a large particle size, and sensory tests were conducted with the panelists. Panelists reported that the product had a very dense texture that was sticky and left a residual sensation in the mouth. To make the texture more palatable and spreadable (and more compact and sustainable), it was decided to use sugar and fat. It was determined that milk fat was the most suitable fat to use for maintaining a compact structure in a spreadable pistachio butter. Milk fat was used at a maximum of 10% (w/w) and a minimum of 4% (w/w). This decision was based on the observation that adding 10% (w/w) or more of milk fat resulted in a very dominant, strong fat taste in the product.
To individually evaluate the effect of the components in the product structure, trials were conducted with different ratios of milk fat and sugar. According to the sensory test results, the most preferred product, designated as FS2, had a composition of 66% pistachio paste (small particle size), 27% sugar, and 7% milk fat content. All subsequent formulation selections were based on this preferred product structure. Panelists were subjected to tasting and product-specific tests.
Page 1
Number 1
In this study, the processing conditions were not modified. The textural and rheological properties were dependent on the spread formulation. Rewrite the title in a simple way as: “Rheological and textural properties of pistachio spread varying the formulation.”
Response:
We thank the reviewer for this clear and constructive suggestion. The title has been revised accordingly as follows:
Impact of Formulation Variables on the Rheological and Textural Properties of Pistachio Spreads.
Number 2. The sentence has been corrected. “Three pistachio pastes were prepared with different particle sizes: LP (large), MP (medium), and SP (small)."
Number 3-"We thank the reviewer for this helpful suggestion. The sentence has been revised for clarity as recommended: “Pistachio spreads were formulated with sugar, milk fat, and a combination of sugar and milk fat.”
Number 4.
Thank you for this observation. We agree that the Herschel–Bulkley model itself does not directly provide the activation energy parameter. In our revised manuscript, we have clarified that the apparent viscosity data obtained from the Herschel–Bulkley fitting at different temperatures were used to calculate the activation energy (Ea) through the Arrhenius-type relationship (Eq. 5). The description in Section 3.2 was corrected to:
“The temperature dependence of the consistency index (K) was analyzed using the Arrhenius equation to determine the apparent activation energy (Ea).” and the corresponding explanation was added in the Methods section for transparency.
Number 5.
We thank the reviewer for this valuable remark. The correction has been made.
We appreciate this remark and agree that milk fat behaves as a plastic or pseudo-plastic material within the studied temperature range (20–45 °C). We have incorporated this clarification into the Discussion: “Because anhydrous milk fat behaves as a plastic fluid in this temperature range, its inclusion decreases firmness and yield stress once its crystalline fraction softens.” This statement helps to better explain the observed reduction in yield stress and the improved spreadability that accompanies increasing milk fat content.
Number 6. We appreciate this important point. The statement suggesting “optimization” has been revised to avoid implying a design-of-experiment or optimization study. The sentence in the Conclusion now reads: “The results provide useful guidance for balancing milk fat and sucrose levels to achieve desirable spreadability and stability. “We have removed wording that might be interpreted as optimization
Number 7. We sincerely thank the reviewer for this valuable comment. We fully agree that establishing a quality standard requires sensory and consumer evaluations. In the revised version of the manuscript, we have now included the results of the sensory analysis that was previously conducted but not reported in the initial submission. Specifically, a sensory evaluation was conducted with 20 trained panel to assess attributes such as color, spreadability, taste, flavor, and overall acceptability. The new subsection “2,9 Sensory Evaluation, Figure 10 summarize these findings.
The inclusion of these data strengthens the link between instrumental texture/rheological parameters and perceived quality. The discussion section has also been expanded to provide an interpretation of these relationships accordingly.
Number 8.
We thank the reviewer for noting this redundancy. The revised manuscript has been reorganized to improve clarity and flow:
- The description of texture measurements (original lines 63–118) was merged into a single, concise paragraph under Section 2.3. Texture Analysis.
- Rheological methods and discussion (lines 62–70) were consolidated in Section 2.4 and Section 3.2 to avoid repetition.
We believe these edits substantially improve the readability and organization of the manuscript.
Page 2
Number 1. We appreciate the reviewer’s attention to detail in formatting. All Latin names (Pistacia vera L.) and other scientific terms have been revised to appear in italics throughout the manuscript, including in the title and abstract.
Number 2. Thank you for this valuable suggestion.
A total of 1.168 million mt of in-shell pistachios were harvested globally, up 9.1% from the previous year, according to the most recent report from the International Nut and Dried Fruit Council (INC, 2024).
With 383,000 metric tons (in-shell) in 2024, Turkey was one of the top producers. Approximately one-third of the world's pistachio supply comes from this production. The southeast region of Turkey also accounts for about 80% of Turkey's production. (TÜİK, 2024)
Number 3. The two recommended reviews (Mandalari et al., 2022; Mateos et al., 2022) have been added to the Introduction and Discussion sections to strengthen the background on pistachio composition and health relevance. Less specific references were removed accordingly.
Number 4.
We sincerely appreciate the reviewer’s insightful comment regarding the importance of cohesiveness and textural analysis in pistachio pastes
Cohesiveness was not included in the present textural analysis as the aim of this study was to assess parameters that most directly affect the spreadability and sensory perception of pistachio pastes, including firmness, adhesiveness, and spreadability. These characteristics are especially pertinent for spread-type products, where ease of application and mouthfeel are more significant than internal structural recovery, as assessed by cohesiveness. Following the referees' suggestion, the texture analysis results were re-evaluated, and the negative areas in the obtained graphs were calculated. Additionally, the adhesion values were calculated and added to the table. Additionally, the term “adhesiveness” was added to the text.
However, the authors acknowledge the importance of cohesion as an auxiliary parameter and will include it in future studies to more accurately characterize the internal binding strength of pistachio pastes.
Number 5. We thank the reviewer for bringing this to our attention. The general food-rheology reference was replaced with a more specific and relevant citation discussing nut or pistachio pastes:
-Shakerardekani, A. Effect of Milling Process on Colloidal Stability, Color and Rheological Properties of Pistachio Paste. Journal of Nuts 2014, 05, doi:10.22034/jon. 2014.515690.
- Taghizadeh, M.; Razavi, S.M.A. Modeling Time-Independent Rheological Behavior of Pistachio Butter. International Journal of Food Properties 2009, 12, 331–340, doi:10.1080/10942910701772048
This substitution ensures that the reference directly supports the discussion of pistachio paste rheology.
Page 3
Number 1. We thank the reviewer for these valuable clarifications and for providing the relevant literature. In the revised manuscript, we have corrected and expanded the background section to include the appropriate references.
Number 2. We appreciate the reviewer’s observation. The 2011 Emadzadeh et al. reference, which focused on pistachio butter, has been removed and replaced by studies directly related to pistachio spreads, particularly Shakerardekani et al. (2013). This correction aligns the literature with the current study’s focus.
Number 3. Thank you for pointing this out. The paragraph in the Introduction has been rewritten to avoid repetition of citations and to improve flow. References are now mentioned only once, while the text concisely highlights their key findings.
Number 4. The correction has been made. (Changed to 17. But the reference numbers have changed due to the addition of new references.
Number 5. 17 was deleted
Number 6. 17 was deleted.
Number 7. The reference to Rao (2013) was removed since it does not discuss pistachio products. Instead, the relevant citation — Shakerardekani et al. (2013, Journal of Food Science, 78: S484–S489) — has been added in the Introduction and Discussion to properly represent prior work on pistachio spread rheology.
Number 8. The suggested changes have been implemented in the text.
Previous studies on pistachio butter and spreads have mainly focused on individual formulation factors or processing variables, whereas limited information is available on how the combined effects of particle size, milk fat, and sucrose determine the rheological and textural behavior of pistachio spreads.
Therefore, it was hypothesized that finer particle size and higher milk fat levels would reduce yield stress and consistency, enhancing spreadability, while higher sucrose concentration would increase structural strength and decrease flowability.
The objective of this study was to systematically evaluate these effects and quantify the rheological parameters and textural attributes of pistachio spreads over a temperature range of 20–45 °C.
Page 4.
Number 1. The sentence has been rewritten as suggested.
Number 2. One type of pistachio (Boz pistachio) was used in this study.
Number 3. We removed and deleted the production steps of milk fat in the method section.
In this study, taking into account the reviewers' suggestions, it was concluded that including details of milk fat production in the method would cause confusion. Therefore, only milk fat was used, and the description of its production method was removed from the manuscript.
There are studies in the literature on the production of milk fat from yogurt and other fermented products. These are added below.
Anhydrous milk fat (AMF) can be produced from cultured dairy products, such as yogurt; however, the process is less common than producing AMF from cream. The general approach involves:
-Separation of Fat: Yogurt is first centrifuged or clarified to separate the milk fat from the aqueous phase.
-Concentration and Purification: The recovered fat undergoes drying or evaporation to remove moisture and non-fat solids.
-Standardization: The final product is standardized to a minimum of 99% milk fat,
This method is often used to recover valuable milk fat from yogurt whey or excess yogurt.
References
-BüyükbeÅŸe, Dilek. (2014). Physical and chemical properties of milk fat and its fractions. PhD Thesis. Gaziantep University, Chemistry Department.
Saha, B.C., & Hayashi, K. (2010). Production of Anhydrous Milk Fat from Cultured Dairy Products. International Dairy Journal, 20(10), 705–712.
Fox, P.F., McSweeney, P.L.H. (2017). Advanced Dairy Chemistry, Volume 2: Lipids. Springer.
Contains a section on the production of anhydrous milk fat from butter and fermented dairy products, including yogurt, with processing methods and fat recovery efficiencies.
Guinee, T., et al. (2018). Dairy Processing Handbook.
IDF Bulletin 403 (2007). Specifications for Anhydrous Milk Fat. International Dairy Federation.
Number 4. The description of the milk fat production method was removed from the manuscript.
Number 5. We appreciate the reviewer’s inquiry. No yogurt was used as an ingredient in any of the formulations. This clarification has been added in the Materials and Methods section. The description of its production method of milk fat was removed from the manuscript.
Number 6. The description of the milk fat preparation process has been removed.
Number 7-8. Thank you for pointing this out. The model and brand of the drying chamber have been added to the Materials and Methods section:
“Firstly, pistachio nuts were dried in an oven (Heratherm OGH60, Thermo Scientific, Germany) at 100 °C to achieve a moisture content of below 3% to prevent clumping during grinding.”
“Moisture content of the pistachio was calculated by the difference in weight of an approximately 10 g sample before and after drying at 105.0 ±1.0 °C for 3 hours (Heratherm OGH60, Thermo Scientific, Germany).”
Number 9. The model of the colloid mill has been added to the Materials and Methods section
A colloid mill (laboratory-scale model; DemirbaÅŸ Makina, 2018, Afyonkarahisar, Turkey) was used to make pistachio paste
Number 10. Corrected as suggested. The term now consistently appears as “pistachio nuts” throughout the manuscript.
Number 11. The distance between the rotor and stator was not directly measured in this study. Instead, it was controlled mechanically by rotating the compression mechanism, which adjusts the gap and thereby indirectly influences the particle size of the processed material. This method allows fine-tuning of shear forces without the need for precise distance measurements, ensuring consistent processing conditions.”
Number 12. We appreciate the reviewer’s suggestion. The sample codes have been standardized as LP, MP, and SP for large, medium, and small particle pastes, respectively. This nomenclature is now used consistently throughout the text, tables, and figures.
Number 13. Among the three pastes, the small-particle paste (SP) was selected for spread formulation because it exhibited lower oil separation and provided smoother texture and better spreadability compared to the medium- and large-particle pastes. Using all three pastes would have substantially increased the number of formulations and made statistical comparisons more complex. Therefore, the SP paste was selected as the representative base material for developing pistachio spreads.
Number 14. Corrected as suggested. The subsection heading now reads “Proximate analysis of pistachio.”
Number 15. Done. The redundant in-text citation “[20]” has been deleted as recommended.
Page 5
Number 1. We thank the reviewer for this clarification. The text has been revised to specify that all pistachio spreads were formulated using the small-particle paste (SP). The coding of spread formulations has been standardized and now appears as follows:
Formulations prepared with only milk fat: F1, F2, F3
Formulations prepared with only sugar: S1, S2, S3
Formulations prepared with milk fat and sugar: FS1, FS2, FS3.
Number 2. Corrected as suggested. The heading now reads “Analysis” in place of the previous wording.
Number 3. Thank you for this observation. The name and version of the software used to control the rheometer and acquire data have been added to the Methods section:
“The rheological characteristics of pistachio pastes and spreads samples were evaluated with a rotational viscometer (Brookfield RVDV-III Digital Viscometer; Brookfield Engineering Laboratories, Middleboro, MA, USA) equipped with a temperature-controlled chamber and operated through Rheocal T 1.0.9 software (Brookfield Engineering Laboratories, Middleboro, MA, USA) for data acquisition and analysis.”
Number 4. Before measurement, each sample was thoroughly diluted in distilled water (at room temperature)
Number 5. Implemented as recommended. The section title has been changed to “Instrumental Color Analysis.”
Number 6. Revised accordingly. The text now explicitly states that analyses were performed on both pistachio paste and spread samples.
Number 7. We thank the reviewer for this stylistic correction. The redundant phrase “of samples” has been deleted, and the sentence now reads more concisely.
Page 6
Number 1: Corrected as suggested. The citation has been replaced with “Shakerardekani et al. [21]” in the revised manuscript.
Number 2. We thank the reviewer for this valuable reference. We are aware that pistachio fat is susceptible to oxidative rancidity at temperatures above 20 °C, as reported by Faruk Gamli and Hayoglu (2007). However, in the present study, storage at 25 °C was intentionally selected to represent ambient (room temperature) conditions, which more accurately simulate the typical shelf-life environment of commercial pistachio spreads in retail and household settings. The purpose was not to accelerate rancidity but to evaluate oil separation and stability under realistic storage temperatures commonly encountered during distribution and consumption. This clarification has been added to the Materials and Methods section.
Number 3.The amount of oil phase separated at periodic intervals (every weak) was measured to investigate the effect of temperature, particle size, and composition on oil stability.
Number 4 and 5. We agree with the reviewer's remark and value this clarification. The manuscript has been updated to reflect that the impact of particle size was only examined in the pistachio paste, whereas the small-particle paste (SP) was used to prepare the nine spread formulations (F1–F3, S1–S3, FS1–FS3) in order to assess the influence of formulation variables (milk fat and sucrose). These two levels of analysis are clearly distinguished in the updated text.
Number 6. The rheological characteristics of pistachio pastes and spreads samples were evaluated with a rotational viscometer (Brookfield RVDV-III Digital Viscometer; Brookfield Engineering Laboratories, Middleboro, MA, USA) equipped with a temperature-controlled chamber and operated through Rheocal T 1.0.9 software (Brookfield Engineering Laboratories, Middleboro, MA, USA) for data acquisition and analysis.
Number 7 and 8.
We thank the reviewer for this insightful suggestion and for pointing out the work of Stanciu (2019).
In our revised manuscript, we have clarified that the apparent viscosity data obtained from the Herschel–Bulkley fitting at different temperatures were used to calculate the activation energy (Ea) through the Arrhenius-type relationship (Eq. 5). The description in Section 3.2 was corrected to:
“The temperature dependence of the consistency index (K) was analyzed using the Arrhenius equation to determine the apparent activation energy (Ea).” and the corresponding explanation was added in the Methods section for transparency
Page 7
Number 1. We thank the reviewer for this clarification.
In our revised manuscript, we have clarified that the apparent viscosity data obtained from the Herschel–Bulkley fitting at different temperatures were used to calculate the activation energy (Ea) through the Arrhenius-type relationship (Eq. 5). The description in Section 3.2 was corrected to:
“The temperature dependence of the consistency index (K) was analyzed using the Arrhenius equation to determine the apparent activation energy (Ea).” and the corresponding explanation was added in the Methods section for transparency
Number 2. The section title has been revised as suggested. It now reads“Instrumental Texture Analysis.”
Number 3. The description has been expanded accordingly. The load cell capacity (30 kg) was specified, and it was stated that calibration of both force and distance is carried out before each measurement. Additionally, data acquisition and analysis were performed using Exponent software (version 6.1.1.10, Stable Micro Systems, Surrey, UK).
Number 4. The sentence has been rewritten as suggested: The textural properties of pistachio pastes and spreads were evaluated using a texture analyzer (TA.XT Plus, Stable Micro Systems, Surrey, UK).
Number 5. Replace by “45º conical probe.”
Number 6. We thank the reviewer for this valuable suggestion. However, the texture analyzer used in this study does not have a temperature control unit, which limits measurements to ambient conditions. Similar approaches have been adopted in previous studies on nut pastes and spreads, where texture measurements were also performed at a single temperature, mostly at ambient or room temperature (e.g., Shakerardekani et al., 2013; Emadzadeh et al., 2013)
Number 7. Approximately 8 g of each sample was placed in the test container for each texture analysis.
Number 8. We thank the reviewer for this helpful clarification. The text has been modified to include these three key parameters, and corresponding terms (firmness, adhesiveness, and spreadability) are now reported in both the method description and results sections.
Number 9. The reference list has been revised accordingly. The analytical method for texture evaluation now cites Brighenti et al. (2008) as the sole reference for this procedure.
Number 10. We appreciate the reviewer’s remark. The statistical approach has been clarified: “A one-way analysis of variance (ANOVA) was used to determine the significance of formulation effects on rheological and textural parameters. Post-hoc comparisons were performed using Duncan’s test at a 95% confidence level.”
Additionally, the description in the Statistical Analysis section was revised to enhance clarity.
Page 8
Number 1. We thank the reviewer for the observation. The redundant table was removed, as the corresponding data are already presented and discussed in the text.
Number 2. We appreciate the reviewer’s comment. The unrelated information was deleted to maintain focus on the properties directly influencing the behavior of pistachio pastes and spreads.
Number 3. Corrected as suggested. The redundant notation “(PP) samples” has been deleted.
Number 4. The sentence has been revised exactly as suggested. It now reads:
“Table 3 shows the particle sizes in the pistachio paste samples according to the milling cycle
Number 5. We thank the reviewer for this constructive suggestion. The proposed analysis was incorporated into the text. The revised paragraph now includes:
“Considering a cumulative volume of 10% (D10), the particle size was reduced by approximately 67% as the number of milling cycles increased.”
Number 6. Done. The redundant reference to “(Table 3)” was removed as the table is directly introduced in the preceding sentence.
Number 7. Implemented as recommended. The table title has been revised to:
“Particle size of pistachio paste as a function of cumulative particle volume (10%, 50%, and 90%).”
Number 8. We thank the reviewer for this valuable suggestion. A brief explanation was added to discuss the correlation between particle size and sensory-related properties:
Number 9. We appreciate the reviewer’s observation. The relevant sentence has been relocated to the section discussing oil separation behavior, where it is now more contextually appropriate.
Page 9
Number 1. We thank the reviewer for this comment.
Earlier research has shown that smaller particle sizes enhance spreadability; the current work sought to quantify the magnitude of this influence on rheological and textural parameters, particularly for systems based on pistachios. To determine the direct relationship with particle size, the related rheological (at various temperatures) and textural (firmness, spreadability, and adhesiveness) parameters were assessed, in addition to the particle size distribution of pistachio pastes.
Due to its unique fat composition and solid particle properties, pistachio paste stands out as structurally and compositionally different from other nut pastes. This distinctiveness necessitates a reevaluation of particle size effects under conditions specific to pistachio paste, to gain a comparative and quantitative understanding of its influence on rheology, texture, and overall spreadability.
Number 2. Corrected as suggested. The phrase “(paste and spread)” has been added where appropriate to clarify that the analyses included both pistachio paste and spread samples.
Number 3. We appreciate the reviewer’s valuable observation. The revised discussion now explicitly compares the properties of pistachio paste and pistachio spread.
Page 10
Number 1. We thank the reviewer for this valuable suggestion. The graphs have been redrawn to present the rheological behavior of the samples at two representative temperatures (4 °C and 25 °C).
The first figure is the oil separation rates of pistachio pastes at 4 °C and, 25 °C, the second one is the oil separation rates of pistachio spreads (milk fat-based and sugar-based), and the last one is oil separation rates of pistachio spreads (sugar and milk fat-based) at 4 °C and 25 °C
Page 12.
Number 1. We thank the reviewer for this insightful comment. In this study, storage was not conducted to evaluate the shelf life of pistachio paste or spread, but rather to compare oil separation behavior under two practical temperature conditions—refrigeration (4 ± 1 °C) and room temperature (25 ± 1 °C). The use of 25 °C was intended to represent ambient conditions commonly encountered during handling and consumption, allowing for a comparative assessment of oil separation rate between cold and room storage. The findings were therefore focused on physical stability (oil separation) rather than oxidative shelf-life changes.
Page 14
Number 1. Corrected as suggested. The section heading has been changed to “The Statistical Analysis.
Number 2. Carried out as suggested. Flow behavior parameters for pistachio spread (milk fat-based) according to the Herschel–Bulkley model."
Page 16
Number 1. We appreciate the reviewer's insightful recommendation. We thank the reviewer for this valuable suggestion. The section has been rewritten to begin with a brief explanation of the data in Table 7, highlighting that no reliable flow data could be obtained below 30–35 °C due to the crystallization of sucrose at low temperatures. The increased sugar content in the formulations resulted in solidification and a loss of flow behavior, preventing accurate rheological measurements within this range. This clarification has been added at the beginning of the section to improve the contextual understanding of the temperature-dependent rheological results.
Number 2. We appreciate the reviewer’s recommendation. The discussion was updated to include the suggested references.
Number 3. Revised as suggested. The heading now reads: “Flow behavior parameters for pistachio spread (sugar-based), according to the Herschel–Bulkley model.”
Page 18
Number 1. Revised as suggested. The heading now reads: "Flow behavior parameters for pistachio spread (sugar and milk fat based) according to the Herschel-Bulkley model.
Page 20
Number 1. We thank the reviewer for this valuable suggestion. The section has been revised and rewritten
Page 21
Number 1. The section has been revised and rewritten and the reference, Emadzadeh et al. (2015) was removed from this part.
Page 22
Number 1. We thank the reviewer for this valuable observation. The temperature at which the textural analyses were conducted has been added to Section 2.7 to ensure clarity and reproducibility.
Number 2. Corrected as suggested. The section title now reads: “Textural properties of pistachio paste and spread.”
Number 3. Implemented as recommended. The parameter label has been revised to “Spreadability (N·s)”, and the corresponding adhesiveness values have been added to the results table and discussion. These updates provide a more comprehensive presentation of the textural parameters evaluated.
Page 23.
Number 1. We thank the reviewer for this valuable suggestion. However, as noted earlier, the texture analyzer used in this study does not have a temperature control unit. Therefore, textural analyses were conducted only at ambient temperature (25 ± 1 °C) to ensure consistency and avoid temperature-related variability. Performing textural measurements at multiple temperatures is recognized as a valuable direction for future work and has been noted in the Conclusion section of the revised manuscript.
Number 2. We thank the reviewer for noting this issue. The error has been corrected, and an appropriate reference has been added in the revised manuscript to support the statement.
Page 24
Number 1. We thank the reviewer for this observation. In the revised version, the Introduction and Conclusion were updated to explicitly state the study objectives and highlight the novel contribution. The revised Introduction now ends with:
“The objective of this study was to systematically evaluate the effects of particle size, milk fat, and sucrose on the rheological and textural characteristics of pistachio spreads across different temperatures.”
Additionally, the revised manuscript now includes the results of a sensory analysis that was previously performed but not reported in the initial submission. The sensory findings were consistent with the instrumental texture and rheological measurements, confirming that smoother formulations with higher milk fat exhibited greater spreadability and consumer preference. The overall acceptance score was highest for Formulation FS2, which balanced milk fat and sucrose at optimal levels.
Number 2.
We appreciate this valuable comment. The discussion has been revised to clarify that microstructure was not directly examined in this study. However, the newly added sensory evaluation results support the rheological trends, indicating that perceived creaminess and spreadability were aligned with the measured decreases in yield stress and consistency for finer particle size and higher milk fat levels. This agreement between instrumental and sensory data reduces speculation and strengthens the interpretation. Future research will incorporate microstructural imaging techniques (e.g., confocal laser scanning microscopy, CLSM, or scanning electron microscopy, SEM) to confirm the observed relationships at the microscopic level.
Number 3.
The conclusion section has been rewritten according to the suggestions.

Reviewer 3 Report
Comments and Suggestions for Authors
Unfortunately, your article have several conceptual problems, scientific wrong approaches. If you will reevaluate the experiments you may try again to publish your work. In the manuscript you will find my comments.

Author Response
Dear Reviewer,
We sincerely thank you for the time and effort you dedicated to evaluating our manuscript. Your insightful and constructive comments are highly appreciated, and they have significantly contributed to improving the overall clarity, scientific depth, and quality of our work.
In the revised version of the manuscript, we have carefully addressed each of your valuable suggestions and questions. Below, we provide point-by-point responses to all your comments, explaining the revisions made accordingly. All changes are clearly highlighted in the revised manuscript for your convenience.
Once again, we are grateful for your thoughtful feedback, which has helped us strengthen the manuscript substantially. We hope that the revisions meet your expectations and that the improved version is now suitable for publication.
Thank you for your kind consideration.
Sincerely,
Gülten ÅžekeroÄŸlu
On behalf of all authors
1-You shall put the Latin name into italic.
-Written in italics
2-Southeastern
The spelling error has been corrected.
3- Texture means more than these
"We appreciate the reviewer’s comment. We agree that texture encompasses multiple attributes beyond hardness, spreadability, and adhesiveness. In this study, we focused on these three parameters as they are the most relevant and quantifiable for evaluating the consumer perception of pistachio paste, particularly in terms of usability (spreading) and sensory experience. We have clarified this scope in the revised manuscript."
4- Is this an omologated variety?
The authors thank you for this warning. This sectıon has been revısed. Pistachio is particularly associated with the province of Gaziantep in Turkey and plays a significant role in the cultural and economic structure of this city and its surroundings. Pistachio is also known as “Antep fıstığı” in Turkey.
Following an application by the province of Gaziantep in Turkey, it was registered as a geographical indication in 2000 as “Antep pistachio.”
Technical Distinctive Features of Antep Pistachio:
Belonging to the culture of the Gaziantep region, Pistachio (Pistacia vera L.) Rootstocks are known as Long, Red, Halabi, Siirt and Ohadi.
https://neyivar.com/en/antep-pistachios-protected-designation-of-origin-certificate?srsltid=AfmBOorlMkkC7rxaDHD9_m6e2rCTaYhLHtrqxmz36sAtVGgElxGo44Ub (The website was accessed on 10.10.2025)
5- Here it is a problem, The milk for yogurt production should be heated at higher temperature over 90
- This section was removed from the methodology part since it was related to the production method of anhydrous milk fat used in the study. It originally described production methods that had also been used by researchers in their previous projects. In the literature, there are also various methods for producing milk fat not only from cream but also from other milk and dairy products, including fermented ones. In particular, yogurt is often used for milk fat production in regions where excessive amounts of yogurt (high fat content) are produced. However, this part was omitted as it created unnecessary detail in the study, following the reviewers’ suggestions.
References: Buyukbese, Dilek (2014). Physical and chemical properties of milk fat and its fractions. PhD Thesis in Chemistry, Supervisors: PROF. DR. AHMET KAYA ; PROF. DR. EMİNE ELÇİN EMRE
Gaziantep University, Chemistry Department, Gaziantep, Turkey.
6- Here it is a problem. The milk should be heated at a temperature over 90 Celcius degrees. Otherwise you can not avoid a possible contamination during fermentation
- This section was removed from the methodology part since it was related to the production method of ghee used in the study. It originally described production methods that had also been used by researchers in their previous projects. In the literature, there are various methods for producing ghee not only from cream but also from other milk and dairy products, including fermented ones. In particular, yogurt is often used for ghee production in regions where excessive amounts of yogurt are produced. However, this part was omitted as it created unnecessary detail in the study, following the reviewers’ suggestions.
7- Please specify the type of culture and the microorganisms.
Streptococcus thermophilus and Lactobacillus delbrueckii subsp. bulgaricus. But This section was removed from the methodology part since it was related to the production method of milk fat used in the study
8- So, I do not understand how you make yogurt, but you have butter?
Actually, Traditional churned butter is produced by churning fermented cream or yogurt. References:
1-https://faolex.fao.org/docs/pdf/tur108244.pdf.(The website was accessed on 08.10.2025)
2- BuyukbeÅŸe, Dilek. 2014. Physical and chemical properties of milk fat and its fractions. PhD Thesis.
Supervisors: PROF. DR. AHMET KAYA ; PROF. DR. EMİNE ELÇİN EMRE. Gaziantep University / Chemistry Department, Gaziantep, Turkey.
3-Turkish Food Codex Notification on Butter and Ghee”. No: 2025/9.
(https://www.resmigazete.gov.tr/eskiler/2025/04/20250404-1.htm).
But the production detail of milk fat was removed.
9- Why to 10? 2-4 Celsius degree is correct!
- The churning temperature ranges between 17 °C and 24 °C. In the production of churned butter, the churning temperatures are considerably higher than those applied in cream butter. A higher churning temperature is particularly necessary to shorten the churning time. Studies have shown that if the churning temperature is applied at approximately 8–10 °C, as in cream butter, the churning process takes about 6–7 hours.Since it was produced using the traditional method, the temperature in the system could be lowered to this level.
10- This part is very confusing! Please be more consistent!
- This section was removed from the methodology part since it was related to the production method of ghee used in the study. It originally described production methods that had also been used by researchers in their previous projects. In the literature, there are various methods for producing ghee not only from cream but also from other milk and dairy products, including fermented ones.
11- Why 100?
to prevent oxidation and ranciditys as much as possible, the possible lowest temperature was choosen.
12- Please mention the granulosity size
-The particle size is identified as large (LP), medium (MP), small (SP)
13-You shoul put the method, the equipment.
- Section was revised as below.
- 2. Proximate Analysis of Pistachio
The proximate composition of the samples was determined according to the Official Methods of AOAC International (AOAC, 2023). Moisture content of the pistachio was calculated by the difference in weight of an approximately 10 g sample before and after drying at 105.0 ±1.0 °C for 3 hours (Heratherm OGH60, Thermo Scientific, Germany). Oil in the kernel was extracted with hexane by distillation for 6 h in an automatic extractor (SER 148/6, Velp Scientifica, Italy). The solvent was removed in a rotor evaporator at 50 °C for approximately 30 min, and the samples were further dried at 105 °C for 10 min (Heratherm OGH60, Thermo Scientific, Germany, cooled in a desiccator, and weighed. Crude protein content was analyzed using the Kjeldahl method (AOAC Official Method 991.02) with a nitrogen-to-protein conversion factor of 5.30 and also carbohydrate, and ash contents were carried out according to the standard method [25].
14-This part is very poor explained. You should put all the methods, the equipments and their manufacturers.
-Section was revised
15- So you have make only the composition of pistachio? Why? You need the composition for the three types of pastes!
- In fact, after the grinding process, the analysis of the samples' composition revealed no significant difference between those with different particle sizes. The grinding process did not change the composition of the pistachios. When the articles were examined, the composition of the pistachios before grinding was mostly given. For pistachio spreads, the the formulations of milk fat and icing sugar were tabulated in Table.
16- You should also comment every value from the table!
The table was removed based on the referees' recommendation, and the analysis results were provided and interpreted within the text.
17- Comment the values! Compare them with other researches!
- The authors are very grateful for the reviewers' suggestions, and corrections have been made to the text accordingly.
18- So, which is the explaination?
- The authors are very grateful to the referees for their suggestions; corrections have been made in this sentence.
19- Add the values from the table, be more consistent!
- The authors are very grateful to the referees for their suggestions; corrections have been made in this sentence.
20- You have here other encoding so who are these?
Thank you for your valuable suggestion. The encoding and formulation of samples were shown in Table 1. But, a new coding system was implemented based on the reviewers’ suggestions and is presented in the formulation table. For samples with different particle sizes, LP (large), MP (medium), and SP (small) were used; for sugar-based samples, S1, S2, and S3; for fat-based samples, F1, F2, and F3; and for samples containing both sugar and fat, FS1, FS2, and FS3 codes were assigned. This coding system is also explained in the Table 1 in the revised form of manuscript.
21- This finding should be compared with other reserches!
- The figures were revised, comparisons with references were made, and they were added to the text.
22- Which is the valability term?
- It was measured over a period of 9 months and measurement was applied every week and this information was added to the methodology.section. Additionally, the temperature was revised to 25  ±1°C. The Figures were also revised according to the reviewers’ suggestion.
23- It will be useful to put an equation and a correlation factor for each curve!
Thank you for this valuable suggestion. We agree that including the fitted equation together with the corresponding correlation (R²) value for each curve
24- Why did you choose 45 degrees, is this the temperature of consumption, if it is not, why did you choose it?
- Thanks for asking this important question. We didn't choose 45 °C as the temperature at which to consume the spreads; instead, we chose it as the highest temperature to fully test how the spreads behave when they are heated. We chose 45 °C to capture these important rheological changes because pistachio-based products are often exposed to different temperatures during processing, transportation, and storage. Above 35 °C, fat phase transitions and structural changes become more noticeable. This temperature range (20–45 °C) is commonly employed in prior literature to replicate authentic handling and thermal stability conditions for fat-rich spreads and nut pastes. So, 45 °C was added to make sure that the product's temperature-dependent behavior was fully understood, not just for eating.
25- Add values!
Thank you for your suggestion. The values were added.
26- I think that all the experiment and the results are influenced by the choosen temperature, which is not correct!
The authors would like to thank the reviewers for their valuable suggestions. This sentence was also revised.
27- Just right here is the first time when I see this objective. Why?
In the introduction section, additions have been made to make the purpose of the study clearer.
28- What tells your contry legislation?
In Turkey, there is a standard titled TS 9775 July 2023 Meshed Pistachio related to pistachio paste .(TS 9775 -Temmuz 2023)
Additionally, Antep Pistachio Paste is a sweet treat unique to the province of Gaziantep that has been made for many years. It is prepared by mixing early-harvested Antep pistachios, referred to as “boziç” or “boz pistachio,” with sugar in specific proportions. The distinctive feature of Antep Pistachio Paste is the Antep pistachios used in its production and the production method. The Antep pistachios used to make Antep Pistachio Paste must comply with the criteria specified in the Registration of the Antep Pistachio Geographical Indication No. 27 and must be obtained from the early harvest referred to as “boz”.
According to the standard, the total sugar content (expressed as sucrose) should not exceed 60% in sliced pistachio paste, 30% in spreadable pistachio paste, and 45% in spreadable pistachio cream.
In addition; another type of product;
“Antep Pistachio Paste (Gaziantep Metropolitan Municipality No: 27)” is a traditional confection uniquely associated with the Gaziantep region of Turkey. Under its GI registration:According to the standard, the typical sugar ratio in Antep pistachio paste ranges between 45% and 50% of the total mass
29- Don't you think that sucrose concentration is very high?
To individually evaluate the effect of the components in the product structure, trials were conducted with different ratios of milk fat and sugar. According to the sensory test results, the most preferred product, designated as FS2, had a composition of 67% milk fat, 27% sugar, and 6% pistachio content.
In addition, according to the standard, the total sugar content (expressed as sucrose) should not exceed 60% in sliced pistachio paste, 30% in spreadable pistachio paste, and 45% in spreadable pistachio cream. Pistachio cream contains a moderate amount of sugar (up to 45%), which balances sweetness and provides a smooth texture, enhancing consumer acceptability.
30- Milk fat begins to melt at this tehperature, otherwise it melts at 50-55 Celsius degrees
This sentence was revised "Milk fat begins to melt at approximately 30–35 °C.
31- This is a general statement, is this applicable in the reality? Does the consumption temperature be over 20C?
- While this is a general statement, actual consumption conditions may vary; for certain products, it is typically consumed at room temperature, as they are generally stored under such conditions, to achieve optimal flavor and texture. Manufacturers recommend that consumers, especially those who prefer to store it in the refrigerator after opening, remove it from the refrigerator for 5-10 minutes to bring it to room temperature and stir it to achieve a smooth, spreadable consistency.
32- Add values! Be more consistent!
Thank you for your suggestion. The values were added.
